# Factorized Tensor Networks for Multi-Task and Multi-Domain Learning

## Abstract

Multi-task and multi-domain learning methods seek to learn multiple tasks/domains, jointly or one after another, using a single unified network. The key challenge and opportunity is to exploit shared information across tasks and domains to improve the efficiency of the unified network. The efficiency can be in terms of accuracy, storage cost, computation, or sample complexity. In this paper, we propose a factorized tensor network (FTN) that can achieve accuracy comparable to independent single-task/domain networks with a small number of additional parameters. FTN uses a frozen backbone network from a source model and incrementally adds task/domain-specific low-rank tensor factors to the shared frozen network. This approach can adapt to a large number of target domains and tasks without catastrophic forgetting. Furthermore, FTN requires a significantly smaller number of task-specific parameters compared to existing methods. We performed experiments on widely used multi-domain and multi-task datasets. We observed that FTN achieves similar accuracy as single-task/domain methods while using 2–6% additional parameters per task. We also demonstrate the effectiveness of FTN with domain adaptation for image generation.

## 1 Introduction

The primary objective in multi-task learning (MTL) is to train a single model to learn multiple related tasks, either jointly or sequentially. Multi-domain learning (MDL) aims to achieve the same learning objective across multiple domains. MTL and MDL techniques seek to improve overall performance by leveraging shared information across multiple tasks and domains. On the other hand, single-task or single-domain learning do not have that opportunity. Furthermore, the storage and computational cost associated with single-task/domain models quickly grows as the number of tasks/domains increases. In contrast, MTL and MDL methods can use the same network resources for multiple tasks/domains, which keeps the overall computational and storage cost small [1–10].

In general, MTL and MDL can have different input/output configurations, but we model them as task/domain-specific network representation problems. Let us represent a network for MTL or MDL as the following general function:

$$\mathbf{y}_t = \mathbf{F}_t(\mathbf{x}) \equiv \mathbf{F}(\mathbf{x}; \mathcal{W}_t, h_t), \tag{1}$$

where $\mathbf{F}_t$ represents a function for task/domain $t$ that maps input $\mathbf{x}$ to output $\mathbf{y}_t$. We further assume that $\mathbf{F}$ represents a network with a fixed architecture and $\mathcal{W}_t$ and $h_t$ represent the parameters for task/domain-specific feature extraction and classification/inference heads, respectively. The function in (1) can represent the network for specific task/domain $t$ using the respective $\mathcal{W}_t, h_t$. In the case of MTL, with $T$ tasks, we can have $T$ outputs $\mathbf{y}_1, \ldots, \mathbf{y}_T$ for a given input $\mathbf{x}$. In the case of MDL, we usually have a single output for a given input, conditioned on the domain $t$. Our main goal is to

Submitted to 37th Conference on Neural Information Processing Systems (NeurIPS 2023). Do not distribute.

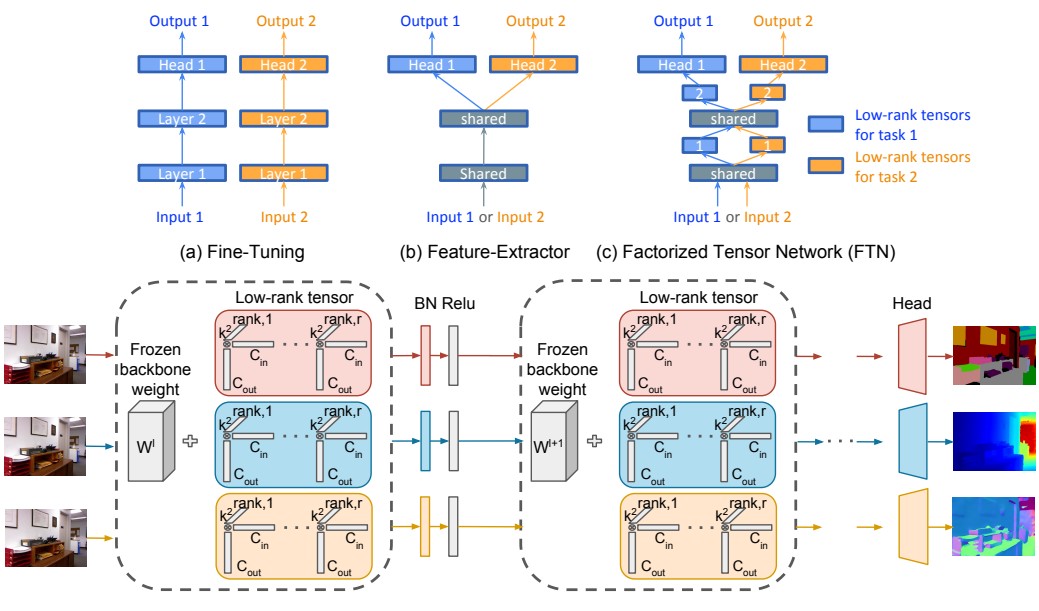

(a) Fine-Tuning    (b) Feature-Extractor    (c) Factorized Tensor Network (FTN)

(d) Detailed overview of FTN architecture.

**Figure 1:** Overview of different MTL/MDL approaches and our proposed method. (a) Fine-Tuning trains entire network per task/domain. (b) Feature-Extractor trains a backbone network shared by all tasks/domains with task/domain-specific heads. (c) Our proposed method, Factorized Tensor Network (FTN), adapts to a new task/domain by adding low-rank factors to shared layers. (d) Detailed overview of FTN. A single network adapted to three downstream vision tasks (segmentation, depth, and surface normal estimation) by adding task-specific low-rank tensors ($\Delta \mathcal{W}_t$). Task/domain-specific blocks are shown in same colors.

learn the $\mathcal{W}_t, h_t$ for all $t$ that maximize the performance of MTL/MDL with minimal computation and memory overhead compared to single task/domain learning.

Figure 1(a),(b),(c) illustrate three typical approaches for MTL/MDL. First, we can start with a pre-trained network and fine-tune all the parameters ($\mathcal{W}_t$) to learn a target task/domain, as shown in Figure 1(a). Fine-Tuning approaches can transfer some knowledge from the pretrained network to the target task/domain, but they effectively use an independent network for every task/domain [1, 5, 11–14]. Second, we can reduce the parameter and computation complexity by using a completely shared Feature-Extractor (i.e., $\mathcal{W}_t = \mathcal{W}_{\text{shared}}$ for all $t$) and learning task/domain-specific heads as last layers, as shown in Figure 1(b). While such approaches reduce the number of parameters, they often result in poor overall performance because of limited network capacity and interference among features for different tasks/domains [1, 4, 5, 15]. Third, we can divide the network into shared and task/domain-specific parameters or pathways, as shown in Figure 1(c). Such an approach can increase the network capacity, provide interference-free paths for task/domain-specific feature extraction, and enable knowledge sharing across the tasks/domains. In recent years, a number of such methods have been proposed for MTL/MDL [1, 4, 5, 9, 16–22]. While existing methods provide performance comparable to single-task/domain learning, they require a significantly large number of parameters.

In this paper, we propose a new method to divide network into shared and task/domain-specific parameters using a factorized tensor network (FTN). In particular, our method learns task/domain-specific low-rank tensor factors and normalization layers. An illustration of our proposed method is shown in Figure 1(d), where we represent network parameters as $\mathcal{W}_t = \mathcal{W}_{\text{shared}} + \Delta \mathcal{W}_t$, where $\Delta \mathcal{W}_t$ is a low-rank tensor. Furthermore, we also learn task/domain-specific normalization parameters. We demonstrate the effectiveness of our method using different MTL and MDL datasets. Our method can achieve accuracy comparable to a single-task/domain network with a small number of additional parameters. Existing parameter-efficient MTL/MDL methods [1, 2, 23] introduce small task/domain-specific parameters while others [15, 24] add many parameters to boost the performance irrespective of the task complexity. In our work, we demonstrate the flexibility of FTNs by selecting the rank according to the complexity of the task. Finally, we also present an experiment for multi-domain image generation using FTNs.

**Contributions.** The main contributions of this paper can be summarized as follows.

- We propose a new method for MTL and MDL, called factorized tensor networks (FTN), that adds task/domain-specific low-rank tensors to shared weights.
- We demonstrate that by using as little as 2–6% additional parameters per task/domain, FTNs can achieve similar performance as the single-task/domain methods.
- Our proposed FTNs can be viewed as a plug-in module that can be added to any pretrained network and layer.
- We performed empirical analysis to show that the FTNs enable flexibility by allowing us to vary the rank of the task-specific tensors based on the complexity of the problem.

**Limitations.** Our proposed method requires a small memory overhead to represent the MTL/MDL networks compared to the single task/domain networks. The proposed method does not affect the computational cost because we need to compute features for each task/domain using separate functional pathways. In our experiments, we used a fixed rank for each layer. In principle, we can adaptively select the rank for different layers to further reduce the parameters. MTL/MDL models often suffer from task interference or negative transfer learning when multiple conflicting tasks are trained jointly. Our method can have similar drawbacks as we did not investigate which tasks/domains should be learned jointly.

## 2 Related Work

**Multi-task learning (MTL)** methods commonly leverage shared and task-specific layers in a unified network to solve related tasks [16–18, 25–32]. These methods learn shared and task-specific representation through their respective modules. Optimization based methods [33–37] devise a principled way to evaluate gradients and losses in multi-task settings. MTL networks that incrementally learn new taks were proposed in [9, 10]. ASTMT [10] proposed a network that emphasizes or suppresses features depending on the task at hand. RCM [9] reparameterizes the convolutional layer into non-trainable and task-specific trainable modules. We compare our proposed method with these incrementally learned networks. Adashare [8] is another related work in MTL that jointly learns multiple tasks. It learns task-specific polices and network pathways [38].

**Multi-domain learning (MDL)** focuses on adapting one network to multiple unseen domains or tasks. MDL setup trains models on task-specific modules built upon the frozen backbone network. This setup helps MDL networks to avoid negative transfer learning or catastrophic forgetting, which is common among multi-task learning methods. The work by [3, 6] introduces the task-specific parameters called residual adapters. The architecture introduces these adapters as a series or parallel connection on the backbone for a downstream task. Inspired by pruning techniques, Packnet [2] learns on multiple domains sequentially on a single task to decrease the overhead storage, which comes at the cost of performance. Similarly, the Piggyback [1] method uses binary masks as the module for task-specific parameters. These masks are applied to the weights of the backbone to adapt them to new domains. To extend this work, WTPB [7] uses the affine transformations of the binary mask on their backbone to extend the flexibility for better learning. BA$^2$ [4] proposed a budget-constrained MDL network that selects the feature channels in the convolutional layer. It gives parameter efficient network by dropping the feature channels based on budget but at the cost of performance. Spot-Tune [24] learns a policy network, which decides whether to pass each image through Fine-Tuning or pre-trained networks. It neglects the parameter efficiency factor and emphasises more on performance. TAPS [5] adaptively learns to change a small number of layers in pre-trained backbone network for the downstream task.

Our proposed method, FTN, achieves performance comparable to or better than other methods by utilizing a fraction of the parameters. We demonstrated that, unlike other methods, easy domains do not require the transformation of the backbone by the same complex module, and we can choose the flexibility of the task-specific parameter for a given domain.

**Domain adaptation and transfer learning.** The work in this field usually focuses on learning a network from a given source domain to a closely related target domain. The target domains under this kind of learning typically have the same category of classes as source domains [11]. Due to this, it benefits from exploiting the labels of source domains to learn about multiple related target domains[12, 39]. Some work has a slight domain shift between source and target data like different

camera views [40]. At the same time, recent papers have worked on large domain shifts like converting targets into sketch or art domains [12, 41]. Transfer learning is related to MDL or domain adaptation, but focuses on how to generalize better on target tasks [13, 14, 42]. Most of the work in this field uses the popular Imagenet as a source dataset to learn feature representation and learn to transfer to target datasets.

# 3 Technical Details

In our proposed method, we use task/domain-specific low-rank tensors to adapt every convolutional layer of a pretrained backbone network to new tasks and domains. Let us assume the backbone network has $L$ convolution layers that are shared across all task/domains. We represent the shared network weights as $\mathcal{W}_{\text{shared}} = \{\mathbf{W}_1, \ldots, \mathbf{W}_L\}$ and the low-rank network updates for task/domain $t$ as $\Delta \mathcal{W}_t = \{\Delta \mathbf{W}_{1,t}, \ldots, \Delta \mathbf{W}_{L,t}\}$. To compute features for task/domain $t$, we update weights at every layer as $\mathcal{W}_{\text{shared}} + \Delta \mathcal{W}_t = \{\mathbf{W}_1 + \Delta \mathbf{W}_{1,t}, \ldots, \mathbf{W}_L + \Delta \mathbf{W}_{L,t}\}$.

To keep our notations simple, let us only consider $l$th convolution layer that has $k \times k$ filters, $C_{in}$ channels for input feature tensor, and $C_{out}$ channels for output feature tensor. We represent the corresponding $\mathbf{W}_l$ as a tensor of size $k^2 \times C_{in} \times C_{out}$. We represent the low-rank tensor update as a summation of $R$ rank-1 tensors as

$$\Delta \mathbf{W}_{l,t} = \sum_{r=1}^{R} \mathbf{w}_{1,t}^r \otimes \mathbf{w}_{2,t}^r \otimes \mathbf{w}_{3,t}^r, \tag{2}$$

where $\mathbf{w}_{1,t}^r, \mathbf{w}_{2,t}^r, \mathbf{w}_{3,t}^r$ represent vectors of length $k^2, C_{in}, C_{out}$, respectively, and $\otimes$ represents the Kronecker product.

Apart from low-rank tensor update, we also optimize over batchnorm layers (BN) for each task/domain [43, 44]. The BN layer learns two vectors $\Gamma$ and $\beta$, each of length $C_{out}$. The BN operation along $C_{out}$ dimension can be defined as element-wise multiplication and addition:

$$\text{BN}_{\Gamma,\beta}(u) = \Gamma \left( \frac{u - \mathbb{E}[u]}{\sqrt{\text{Var}[u] + \epsilon}} \right) + \beta. \tag{3}$$

We represent the output of $l$th convolution layer for task/domain $t$ as

$$\mathbf{Z}_{l,t} = \text{BN}_{\Gamma_t, \beta_t}(\text{conv}(\mathbf{W}_l + \Delta \mathbf{W}_{l,t}, \mathbf{Y}_{l-1,t})), \tag{4}$$

where $\mathbf{Y}_{l-1,t}$ represents the input tensor and $\mathbf{Z}_{l,t}$ represents the output tensor for $l$th layer. In our proposed FTN, we learn the task/domain-specific factors $\{\mathbf{w}_{1,t}^r, \mathbf{w}_{2,t}^r, \mathbf{w}_{3,t}^r\}_{r=1}^{R}$, and $\Gamma_t$, and $\beta_t$ for every layer in the backbone network.

In the FTN method, since we are learning over only $\Delta \mathbf{W}$ and BN parameters, the rank, $R$, plays an important role in defining the expressivity of our network. We can define a complex $\Delta \mathbf{W}$ by increasing the rank $R$ of the low-rank tensor and taking their linear combination. Our experiments showed that this has resulted in a significant performance gain.

**Initialization.** In our approach, the initialization of the low-rank parameter layers and the pre-trained weights of the backbone network plays a crucial role due to their sensitivity towards performance. To establish a favorable starting point, we adopt a strategy that minimizes substantial modifications to the frozen backbone network weights during the initialization of the task-specific parameter layers. To achieve this, we initialize each parameter layer from the xavier uniform distribution [45], thereby generating $\Delta \mathbf{W}$ values close to 0 before their addition to the frozen weights. This approach ensures the maintenance of a similar initialization state to the frozen weights at iteration 0.

To acquire an effective initialization for our backbone network, we leverage the pre-trained weights obtained from ImageNet. We aim to establish a robust and capable feature extractor for our specific task by incorporating these pre-trained weights into our backbone network.

**Number of parameters.** In a Fine-Tuning setup with $T$ tasks/domains, the total number of required parameters at convolutional layer $l$ can be calculated as $T \cdot (k^2 \times C_{in} \times C_{out})$. Whereas using our proposed FTNs, the total number of frozen backbone ($\mathbf{W}_l$) and low-rank R tensor ($\Delta \mathbf{W}_{l,t}$) parameters are given by $(C_{out} \times C_{in} \times k^2) + T \cdot R \cdot (C_{out} + C_{in} + k^2)$. In our results section, we

**Table 1:** Number of parameters and top-1% accuracy for baseline methods, comparative methods, and FTN with varying ranks on the five domains of the ImageNet-to-Sketch benchmark experiments. Additionally, the mean top-1% of each method across all domains is shown. The 'Params' column gives the number of parameters used as a multiplier of those for the Feature-Extractor method, along with the absolute number of parameters required in parentheses.

| Methods | Params (Abs) | Flowers | Wikiart | Sketch | Cars | CUB | mean |
|---|---|---|---|---|---|---|---|
| Fine-Tuning | $6\times$ (141M) | 95.69 | 78.42 | 81.02 | 91.44 | 83.37 | 85.98 |
| Feature-Extractor | $1\times$ (23.5M) | 89.57 | 57.7 | 57.07 | 54.01 | 67.20 | 65.11 |
| FC and BN only | $1.001\times$ (23.52M) | 94.39 | 70.62 | 79.15 | 85.20 | 78.68 | 81.60 |
| Piggyback [1] | $6\times$ [$2.25\times$] (141M) | 94.76 | 71.33 | 79.91 | 89.62 | 81.59 | 83.44 |
| Packnet $\rightarrow$ [2] | [$1.60\times$] (37.6M) | 93 | 69.4 | 76.20 | 86.10 | 80.40 | 81.02 |
| Packnet $\leftarrow$ [2] | [$1.60\times$] (37.6M) | 90.60 | 70.3 | 78.7 | 80.0 | 71.4 | 78.2 |
| Spot-Tune [24] | $7\times$ [$7\times$] (164.5M) | 96.34 | 75.77 | 80.2 | 92.4 | 84.03 | 85.74 |
| WTPB [7] | $6\times$ [$2.25\times$] (141M) | 96.50 | 74.8 | 80.2 | 91.5 | 82.6 | 85.12 |
| BA$^2$ [4] | $3.8\times$ [$1.71\times$] (89.3M) | 95.74 | 72.32 | 79.28 | 92.14 | 81.19 | 84.13 |
| TAPS [5] | $4.12\times$ (96.82M) | 96.68 | 76.94 | 80.74 | 89.76 | 82.65 | 85.35 |
| **FTN, R=1** | **$1.004\times$** (23.95M) | 94.79 | 73.03 | 78.62 | 86.85 | 80.86 | 82.83 |
| **FTN, R=50** | $1.53\times$ (36.02M) | 96.42 | 78.01 | 80.6 | 90.83 | 82.96 | **85.76** |

have shown that the absolute number of parameters required by our method is a fraction of what the Fine-Tuning counterpart needs.

**Effect of batch normalization.** In our experiment section, under the 'FC and BN only' setup, we have shown that having task-specific batchnorm layers in the backbone network significantly affects the performance of a downstream task/domain. For all the experiments with our proposed approach, we include batch normalization layers as task-specific along with low-rank tensors and classification/decoder layer.

## 4 Experiments and Results

We evaluated the performance of our proposed FTN on several MTL/MDL datasets for three different experiments: **1. Multi-domain classification**, **2. Multi-task dense prediction**, and **3. Multi-domain image generation**. For each set of benchmarks we report the performance of FTN with different rank increments and compare with results from existing methods. All experiments are run on a single NVIDIA GeForce RTX 2080 Ti GPU with 12GB memory.

### 4.1 Multi-domain classification

**Datasets.** We use two MTL/MDL classification-based benchmark datasets. First, ImageNet-to-Sketch [1, 2, 5, 7], which contains five different domains: Flowers [46], Cars [47], Sketch [48], Caltech-UCSD Birds (CUBs) [49], and WikiArt [50], with different classes. Second, DomainNet [51], which contains six domains: Clipart, Sketch, Painting (Paint), Quickdraw (Quick), Inforgraph (Info), and Real, with each domain containing an equal 345 classes. The datasets are prepared using augmentation techniques as adopted by [5].

**Training details.** For each benchmark, we report the performance of FTN for various choices for ranks, along with several benchmark-specific comparative and baseline methods. The backbone weights are pretrained from ImageNet, using ResNet-50 [52] for the ImageNet-to-Sketch benchmarks, and ResNet-34 on the DomainNet benchmarks to keep the same setting as [5]. On ImageNet-to-Sketch we run FTNs for ranks, $R \in \{1, 5, 10, 15, 20, 25, 50\}$ and on DomainNet dataset for ranks, $R \in \{1, 5, 10, 20, 30, 40\}$. In the supplementary material, we provide the hyperparameter details to train our network.

**Results.** We report the top-1% accuracy for each domain and the mean accuracy across all domains for each collection of benchmark experiments. We also report the number of frozen and learnable parameters in the backbone network. Table 1 compares the FTN method with other methods in terms of accuracy and number of parameters (also see Figure 2). FTN outperforms every other method while using 36.02 million parameters in the backbone with rank-50 updates for all domains. The

**Table 2:** Performance of different methods with resnet-34 backbone on DomainNet dataset. Top-1% accuracy is shown on different domains with different methods along with the number of parameters.

| Methods | Params (Abs) | Clipart | Sketch | Paint | Quick | Info | Real | mean |
|---|---|---|---|---|---|---|---|---|
| Fine-Tuning | 6× (127.68M) | 74.26 | 67.33 | 67.11 | 72.43 | 40.11 | 80.36 | 66.93 |
| Feature-Extractor | 1× (21.28M) | 60.94 | 50.03 | 60.22 | 54.01 | 26.19 | 76.79 | 54.69 |
| FC and BN only | 1.004× (21.35M) | 70.24 | 61.10 | 64.22 | 63.09 | 34.76 | 78.61 | 62.00 |
| Adashare [8] | 5.73× (121.93M) | 74.45 | 64.15 | 65.74 | 68.15 | 34.11 | 79.39 | 64.33 |
| TAPS [5] | 4.90× (104.27M) | 74.85 | 66.66 | 67.28 | 71.79 | 38.21 | 80.28 | **66.51** |
| **FTN, R=1** | **1.008**× (21.44M) | 70.73 | 62.69 | 65.08 | 64.81 | 35.78 | 79.12 | 63.03 |
| **FTN, R=40** | 1.18× (25.22M) | 74.2 | 65.67 | 67.14 | 71.00 | 39.10 | 80.64 | 66.29 |

mean accuracy performance is better than other methods and is close to Spot-Tune [24], which requires nearly 165M parameters. On the Wikiart dataset, we outperform the top-1 accuracy with other baseline methods. The performance of baseline methods is taken from TAPS [5] since we are running the experiments under the same settings.

Table 2 shows the results on the DomainNet dataset, which we compare with TAPS [5] and Adashare [8]. Again, using FTN, we significantly outperform comparison methods along the required parameters (rank-40 needs 25.22 million parameters only). Also, FTN rank-40 attains better top-1% accuracy on the Infograph and Real domain, while it attains similar performance on all other domains. On DomainNet with resnet-34 and Imagenet-to-Sketch with resnet-50 backbone, the rank-1 low-rank tensors require only 16,291 and 49,204 parameters per task, respectively. We have shown additional experiments on this dataset under a joint optimization setup in the supplementary material.

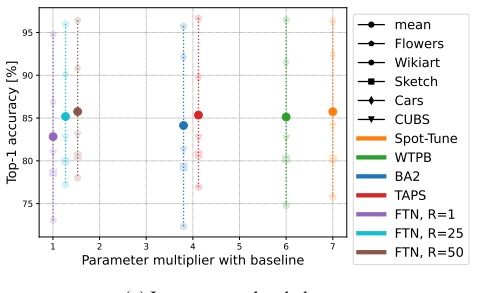
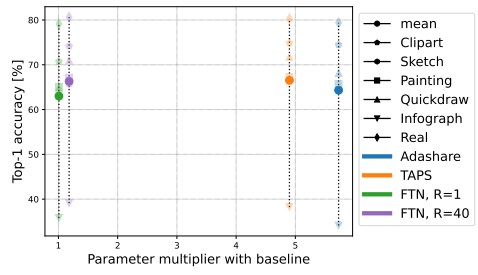

**(a)** Imagenet-to-sketch dataset      **(b)** DomainNet dataset

**Figure 2: Accuracy vs Parameter multiplier with baseline:** We show the top-1% accuracy against the number of parameter increments through our approach in the backbone network with the baseline backbone. We plot the performance of our method with other baseline methods, which has shown that our approach attains competitive performance with an extremely small number of parameters.

**Analysis on rank.** We showed the effect of rank on FTNs by performing experiments with multiple ranks on both datasets. Figure 3 shows the accuracy vs. ranks plot, where we observe a trend of performance improvement as we increase the rank from 1 to 50 on the ImageNet-to-Sketch and from 1 to 40 on the DomainNet dataset. Also, not all domains need a high rank, as Figure 3 shows that the Flowers and Cars domain attains good accuracy at rank 20 and 15, respectively. We can argue that, unlike prior works [24, 23], which consume the same task-specific module for easy and complex tasks, we can provide different flexibility to each task. Also, in supplementary material, we have a heatmap plot showing the adaption of low-rank tensor at every layer upon increasing the rank.

## 4.2  Multi-task dense prediction

**Dataset.** The widely-used NYUD dataset [53] with 795 training and 654 testing images of indoor scenes is used for dense prediction experiments in multi-task learning. The dataset contains four tasks: edge detection (Edge), semantic segmentation (SemSeg), surface normals estimation (Normals), and depth estimation (Depth). We follow the same data-augmentation technique as used by [9].

**Metrics.** On the tasks of the NYUD dataset, we report mean intersection over union for semantic segmentation, mean error for surface normal estimation, optimal dataset F-measure [54] for edge

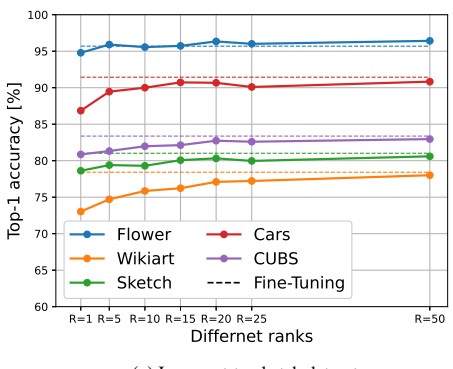

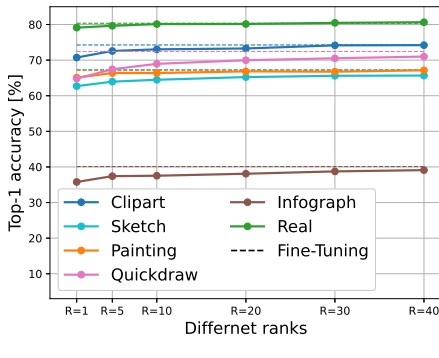

| (a) Imagenet-to-sketch dataset | (b) DomainNet dataset |

**Figure 3: Accuracy vs Low-ranks:** We show the top-1% accuracy against the different low-ranks used in our method for different domains. We start with an 'only BN' setup where without any low-rank we keep the BN (batchnorm) layers as task-specific. Then we show the performance improvement through our approach upon increasing the rank, R.

detection, and root mean squared error for depth estimation. We also report the number of parameters used in the backbone for each method.

**Training details.** ResNet-18 is used as the backbone network, and DeepLabv3+ [55] as the decoder architecture. The Fine-Tuning and Feature-Extractor experiments are implemented in the same way as in the classification-based experiments above. We showed experiments for FTNs with $R \in \{1, 10, 20, 30\}$. Further details are in the supplementary material.

**Results.** Table 3 shows the performance of FTN with various ranks and of other baseline comparison methods for dense prediction tasks on the NYUD dataset. We observe performance improvement by increasing flexibility through higher rank. FTN with rank-30 performs better than all comparison methods and utilizes the least number of parameters. Also, on the Depth and Edge task we can attain good performance by using only rank-20. We take the performance of baseline comparison methods from the RCM paper [9] as we run our experiments under the same setting.

**Table 3:** Dense prediction performance on NYUD dataset using ResNet-18 backbone with DeepLabv3+ decoder. The proposed FTN approach with $R = \{1, 10, 20, 30\}$ and other methods. The best performing method in bold.

| Methods | Params (Abs) | Semseg↑ | Depth↓ | Normals↓ | Edge↑ |
|---|---|---|---|---|---|
| Single Task | 4× (44.68M) | 35.34 | 0.56 | 22.20 | 73.5 |
| Decoder only | 1× (11.17M) | 24.84 | 0.71 | 28.56 | 71.3 |
| Decoder + BN only | 1.002× (11.19M) | 29.26 | 0.61 | 24.82 | 71.3 |
| ASTMT (R-18 w/o SE) [10] | 1.25× (13.99M) | 30.69 | 0.60 | 23.94 | 68.60 |
| ASTMT (R-26 w SE) [10] | 2.00× (22.42M) | 30.07 | 0.63 | 24.32 | 73.50 |
| Series RA [3] | 1.56× (17.51M) | 31.87 | 0.60 | 23.35 | 67.56 |
| Parallel RA [6] | 1.50× (16.77M) | 32.13 | 0.59 | 23.20 | 68.02 |
| RCM [9] | 1.56× (17.49M) | 34.20 | 0.57 | 22.41 | 68.44 |
| **FTN, R=1** | **1.005× (11.23M)** | 29.83 | 0.60 | 23.56 | 72.7 |
| **FTN, R=10** | 1.03× (11.54M) | 33.66 | 0.57 | 22.15 | 73.5 |
| **FTN, R=20** | 1.06× (11.89M) | 34.06 | **0.55** | 21.84 | **73.9** |
| **FTN, R=30** | 1.09× (12.24M) | **35.46** | 0.56 | **21.78** | 73.8 |

### 4.3 Multi-domain image generation

A deep generative network $\mathbf{G}$, parameterized by $W$, can learn to map a low-dimensional latent code $\mathbf{z}$ to a high-dimensional natural image $\mathbf{x}$ [56–58]. To find a latent representation for a set of images given a pre-trained generative network, we can solve the following optimization problem:

$$\min_{z_i} \sum_{i=1}^{N} \|x_i - \mathbf{G}(z_i; \mathcal{W})\|_p^p. \tag{5}$$

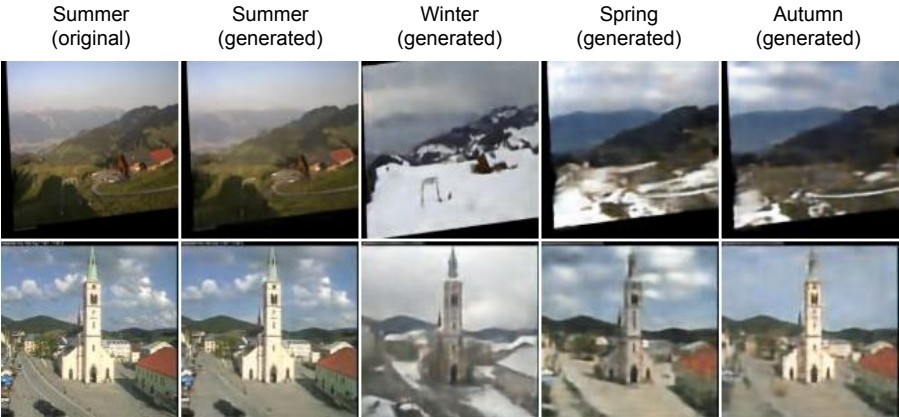

**Figure 4:** Generated images for different seasons using FTN.

The work in [58] as well as our experimental results show that this approach is very limited in handling complex and diverse images. If $\mathbf{x}$ is an image that belongs to a domain that is different from the source domain used to train the generator, we are not guaranteed to find a latent vector $\mathbf{z}^*$ such that $\mathbf{x} \approx \mathbf{G}(\mathbf{z}^*)$. We showed FTNs can be used to expand the range of $\mathbf{G}$ by reparametrizing it as $\mathbf{G}(z_i; \mathcal{W}, \Delta\mathcal{W}_t)$, where $\Delta\mathcal{W}_t$ are the domain-specific low-rank factors and Batch Normalization parameters. By optimizing over the latent vectors $z_i$ and domain specific parameters, we learned to generate images from new domains.

**Dataset.** We used the multi-domain Transient Attributes [59] dataset that contains outdoor scenes under different weather, lighting, and seasons. We extracted season information, "summer", "winter", "spring", and "autumn", for each image and categorized them accordingly. Our goal is to learn a single FTN network that can generate images from all seasons.

**Training details.** Our base network follows the BigGAN architecture [60]. We compared our proposed FTN network with models trained under two different setups. In the first setup, which serves as our baseline, we fine-tuned a pre-trained generator on images from all seasons. For the second setup, we fine-tuned the same network separately for each season. Additional training details can be found in the supplementary material.

**Results.** Table 4 shows the performance, in terms of Peak Signal-to-Noise Ratio (PSNR), for three methods. The baseline model is a single network trained on all images and shows overall poor performance. Our proposed FTN network achieved a comparable performance to the single-domain networks. Each single domain network has 71.4M trainable parameters, while the FTN network adds an additional 3.9M

**Table 4:** Image generation PSNR for different methods and seasons

| Season | Baseline | Single domain | FTN |
|--------|----------|---------------|------|
| Summer | 10.80 | 25.30 | 25.30 |
| Winter | 9.24 | 21.30 | 22.23 |
| Spring | 11.08 | 22.07 | 20.50 |
| Autumn | 10.92 | 19.87 | 20.08 |

parameters per domain over the base network. Figure 4 shows examples of images generated by our proposed FTN network.

## 5 Conclusion

We have proposed a simple, architecture-agnostic, and easy-to-implement FTN method that adapts to new unseen domains/tasks by using low-rank task-specific tensors. In our work, we have shown FTN requires the least number of parameters than other baseline methods in MDL/MTL experiments and attains better or comparable performance. We can adapt the backbone network with different flexibility using low-ranks based on the complexity of the domain/task. We conducted experiments with different backbone architectures, and our work can be extended to transformer-based architecture. Furthermore, we demonstrate experiments with FTN on image generation.

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
