# Factorized Tensor Networks for Multi-Task and Multi-Domain Learning
# (Supplementary Material)

1 In this section, we present additional material and details to supplement the main paper.

## A  Dataset and training details

### A.1  Imagenet-to-sketch dataset

The dataset contains five different domains: Flowers [1], Cars [2], Sketch [3], Caltech-UCSD Birds (CUBs) [4], and WikiArt [5] with 102, 195, 250, 196, and 200 classes, respectively. We randomly crop images from each domain to $224 \times 224$ pixels, along with normalization and random horizontal flipping. We report the baseline experiments, Fine-Tuning, Feature-Extractor, and 'FC and BN only' with 0.005 learning rate (lr) and SGD optimizer with no weight decay. We train for 30 epochs with a batch size 32 and a cosine annealing learning rate scheduler.

The experiments with our proposed FTN approach are learned through the Adam optimizer with lr 0.005 for low-rank layers and through the SGD optimizer with lr 0.008 for remaining trainable layers (task-specific batchnorm and classification layers). Again, we train them for 30 epochs with batch size 32 and cosine annealing scheduler. We showed the experiments for different low-ranks, $R \in \{1, 5, 10, 15, 20, 25, 50\}$.

### A.2  DomainNet dataset

This dataset contains six domains: Clipart, Sketch, Painting (Paint), Quickdraw (Quick), Inforgraph (Info), and Real, with an equal number of 345 classes/categories. We train the baseline experiments, Fine-Tuning, Feature-Extractor, and 'FC and BN only' with 0.005 lr with 0.0001 weight decay and SGD optimizer. Similar to the Imagenet-to-sketch dataset, we apply the same data augmentation techniques and train for 30 epochs with 32 batch size and a cosine annealing learning rate scheduler.

For our experiments with the FTN method, we train the low-rank tensor layers with Adam optimizer and 0.005 lr. The remaining layers were optimized using the SDG optimizer with the same 0.005 learning rate and no weight decay. We train the FTN networks for 30 epochs with the same learning rate scheduler. We showed our experiments for different low-ranks, $R \in \{1, 5, 10, 20, 30, 40\}$.

### A.3  NYUD dataset

In multi-task learning, we use NYUD dataset, which consists of 795 training and 654 testing images of indoor scenes, for dense prediction experiments. It has four tasks: edge detection (Edge), semantic segmentation (SemSeg), surface normals estimation (Normals), and depth estimation (Depth). We evaluated the performance using optimal dataset F-measure (odsF) for edge detection, mean intersection over union (mIoU) for semantic segmentation, and mean error (mErr) for surface normals. At the same time, we report root mean squared error (RMSE) for depth. We perform random

scaling in the range of [0.5, 2.0] and random horizontal flipping for data augmentation and resize each image to $425 \times 560$. We train our baseline experiments, Fine-tuning, Feature Extractor, and 'FC and BN only' for 60 epochs with batch size 8 and polynomial learning rate scheduler. We learn the network using SGD optimizer and 0.005 learning rate with 0.9 momentum and 0.0001 weight decay.

In FTN we train for the same 60 epochs, batch size 8, and polynomial learning rate scheduler. We learn over low-rank layers using the Adam optimizer with a 0.01 learning rate and no weight decay. The remaining decoder and batchnorm layers are optimized using the same hyperparameters used for baseline experiments. This dataset shows experiments with different low-ranks, $R \in \{1, 10, 20, 30\}$.

## B  Effect on performance with different number of low-rank factors.

We performed an experiment by removing the low-rank factors from our trained FTN backbone network at different thresholds. We perform this experiment on five domains of the Imagenet-to-sketch dataset and compute the $\ell_2$-norm of $\Delta\mathbf{W}$ at every layer. We selected equally spaced threshold values from the minimum and maximum $\ell_2$-norm and removed the low-rank factors below the threshold. The performance vs. the number of parameters of low-rank layers for different thresholds is shown in Figure S1. We observe a drop in performance on every domain as we increase the threshold and reduce the number of layers from the backbone. Interestingly, when we reduce the number of layers from 52 to 28 on the Flowers and CUBS dataset, we did not see a significant drop in accuracy.

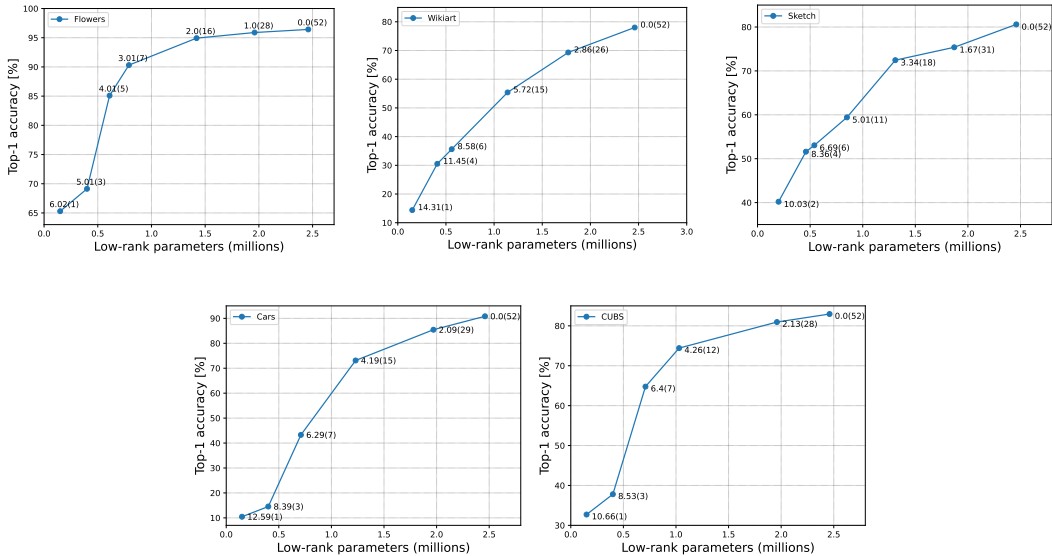

**Figure S1:** Performance on five domains of the Imagenet-to-sketch dataset as we remove the low-rank parameters. We selected the number of layers in the backbone based on a moving threshold. We annotate the specified threshold at each marker point and the number of affected layers (in parentheses).

## C  Visualization of changes in FTN with low-rank factors

We present the norm of low-rank factors at every adapted layer in the backbone of our FTN as a heatmap in Figures S2–S3. The colors indicate relative norms because we normalized them for every network in the range 0 to 1 to highlight the relative differences. Figure S2 presents results on five domains of the Imagenet-to-sketch dataset (resnet-50 backbone), adapting every layer with rank-50 FTN. We observe the maximum changes in the last layer of the backbone network instead of the initial layers. We also show a similar trend on the DomainNet dataset with resnet-34 backbone where maximum changes occur in the network's later layer (see Figure S4). We observe from Figure S3 that on the wikiart domain of the Imagenet-to-sketch dataset, the layers in the backbone network become more adaptive upon increasing the rank of FTN. The rank-50 FTN has more task adaptive layers than the rank-1 FTN on the wikiart domain.

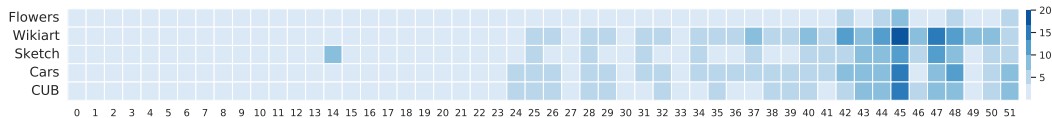

**Figure S2:** Norm of low-rank factors in the adapted backbone layers for different domains of the Imagenet-to-sketch dataset with $R = 50$.

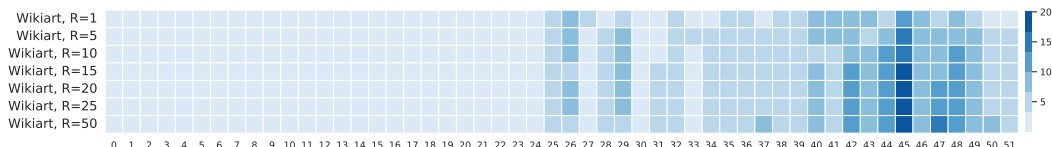

**Figure S3:** Norm of low-rank factors in the adapted backbone layers for different values of $R \in \{1, 5, 10, 15, 20, 25, 50\}$ with the wikiart domain of the Imagenet-to-sketch dataset.

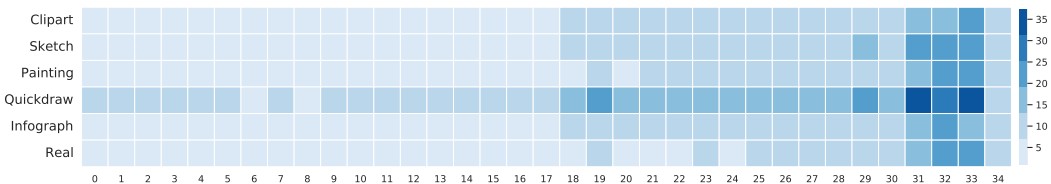

**Figure S4:** Norm of low-rank factors in the adapted backbone layers for different domains of the DomainNet dataset with $R = 40$.

# D Results on DomainNet dataset under joint setting

We performed additional experiments under a joint learning setup on the DomainNet dataset. A single model is trained jointly on all the domains of the dataset with Fine-Tuning setup. Table S1 summarizes the results for the performance of the DomainNet dataset under a joint setup. The domains under this setting share information among each other for all parameters. The Fine-Tuning experiment achieves the overall best performance, but at the expense of a large number of parameters. We observe poor performance for the shared Feature Extractor, since that does not learn any additional task-specific parameters. The results in the third row show that by changing just the task-specific batchnorm layers in the jointly trained backbone, we can achieve better results than TAPS and Adashare. Our FTN with just $R = 1$ also outperforms Adashare and TAPS.

**Table S1:** Performance on DomainNet dataset under joint setting using resnet34 backbone (initialized with jointly trained weights) with our FTN approach along with comparison methods.

| Methods | Params (Abs) | Clipart | Sketch | Paint | Quick | Info | Real | mean |
|---|---|---|---|---|---|---|---|---|
| Fine-tuning | $6\times$ (127.68M) | 77.43 | 69.25 | 69.21 | 71.61 | 41.50 | 80.74 | 68.29 |
| Feature extractor | $1\times$ (21.28M) | 76.67 | 65.2 | 65.26 | 52.97 | 35.05 | 76.08 | 61.87 |
| FC and BN only | $1.004\times$ (21.35M) | 77.07 | 68.34 | 68.76 | 69.06 | 40.63 | 79.07 | 67.15 |
| Adashare | $1\times$ (21.28M) | 75.88 | 63.96 | 67.90 | 61.17 | 31.52 | 76.90 | 62.88 |
| TAPS | $1.46\times$ (31.06M) | 76.98 | 67.81 | 67.91 | 70.18 | 39.30 | 78.91 | 66.84 |
| **FTN, R=1** | $1.008\times$ (21.45M) | 77.13 | 68.10 | 68.50 | 69.41 | 40.04 | 79.49 | 67.11 |

# E Image generation training details and results

## E.1 Transient attributes dataset

The Transient attributes dataset [6] contains a total of 8571 images with 40 annotated attribute labels. Each label is associated with a score in the $[-1, 1]$ range. We utilize the associated confidence score for each season to build our collection of images for each season. Additionally, we only selected

**Table S2:** FTN PSNR for image generation under $R = \{20, 50\}$

| Season | Rank 20 | Rank 50 |
|--------|---------|---------|
| Winter | 18.11 | 22.23 |
| Spring | 18.89 | 20.50 |
| Autumn | 17.95 | 20.08 |

images that were captured during daytime. Our training set consisted of 1875 summer, 1405 spring, 1353 autumn, and 2566 winter images. We normalized each image to the range of $[-1, 1]$ and resized them to a resolution of $128 \times 128$.

### E.2 Training details

Our base network follows the BigGAN architecture [7] that was pre-trained for 100k iterations on ImageNet using $128 \times 128$ images. We trained all the networks in this experiment using Adam optimizer. We used a learning rate of 0.05 for the low-rank tensors and a learning rate of 0.001 for the linear layers. We did not use any weight decay. We trained for 2000 epochs with a cosine annealing learning rate scheduler and an early stopping criterion ranging from 200 to 600 iterations.

### E.3 Additional results

Figure S5, we show additional generated images by our FTN network. In addition, table S2 shows the average performance of our FTN network under different rank settings. We observe a performance increase by increasing the rank $R$ of our low-rank factors.

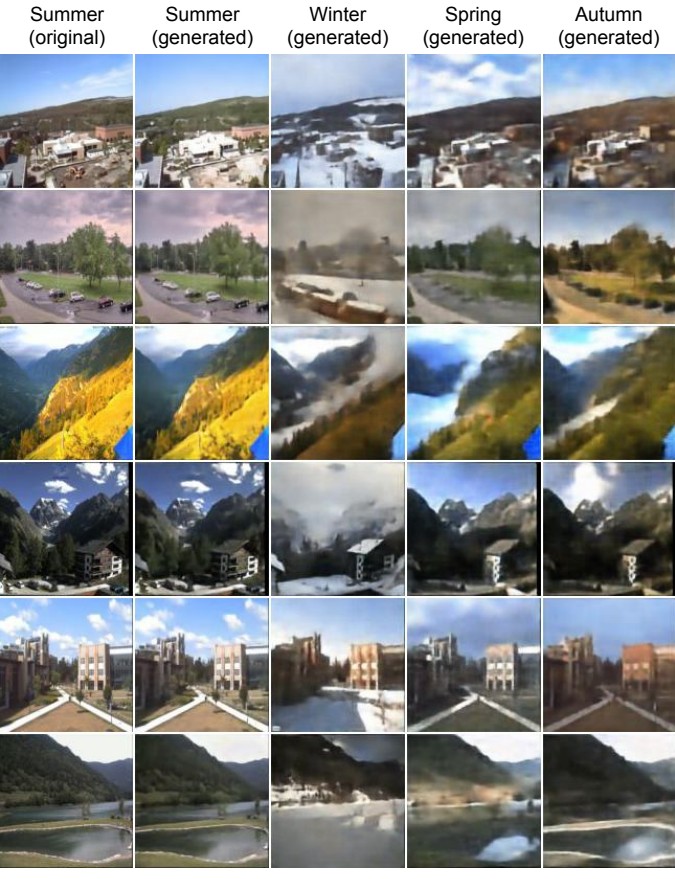

**Figure S5:** Generated images for different seasons using FTN.