# OpenReview forum: "Factorized Tensor Networks for Multi-Task and Multi-Domain Learning"
_NeurIPS.cc/2023/Conference — Submitted to NeurIPS 2023_

### Official Review · Reviewer_TB3s · 2023-06-18

**Soundness:** 3 good
**Presentation:** 3 good
**Contribution:** 3 good
**Rating:** 6
**Confidence:** 5

**Summary:**

This paper proposes a factorized tensor network (FTN), that adds task/domain-specific low-rank tensors to shared weights, to perform lightweight MTL/MDL. Specifically, the low-rank tensors can form a weight additive to the shared backbone weight so that each task can have its own parameters to achieve good accuracy.

**Strengths:**

- The paper is easy to follow.
- The basic idea of adding additional task-specific low-rank vectors is clear to understand.

**Weaknesses:**

- About the method:
    - My high-level concern about this idea is not truly an MTL/MDL work in my understanding.
    If I understand the proposed method correctly, it is equivalent to fine-tuning a completely shared feature extractor on each task. After fine-tuning, we can get the weights offset $\Delta W = W_t - W$ between the fine-tuned weights $W_t$ for task $t$ and the original backbone weights $W$. The low-rank tensors are just a parameter-efficient way to store $\Delta W$ by three separate vectors.
    What I believe is that the proposed method is still a traditional transfer learning method with some low-rank storage design for the weight difference.

    - If it is an MTL work, it fails to obtain the two basic advantages of using MTL.
        - First, in MTL, we believe a good parameters-sharing may improve the generalization ability, i.e., the task accuracy, when training multiple tasks jointly. However, in this paper, the shared backbone weights will be frozen (as shown in Figure 1d) when training the additional low-rank tensors for each task, which means there is no truly joint training/parameter sharing across the multiple tasks. As I described in the first bullet point, each task is just fine-tuned from the shared backbone and has no communication with other tasks during training, and the final model weights for each task will be independent to each other at the end.
        - Second, the wide adaptation of MTL relies on its advantage of lower computation cost and latency. However, the proposed method in this paper has no latency saving since all the tasks still have their own parameters and have to execute sequentially even if they share the same input images.

    - About the method itself, I feel it lacks good motivation. There is no reason why we can have three low-rank tensors to get $\Delta W$ for each task in each layer. The authors may consider adding some literature that can support this choice, or have some ablation studies. For example, what will happen if we use the full-rank $\Delta W$ directly? How about using PCA or SVD on $\Delta W$ to get low-rank matrics?

    - More importantly, this paper has a very similar idea as [1]. The only difference is the choice of low-rank representation.
    [1] Hu E J, Shen Y, Wallis P, et al. Lora: Low-rank adaptation of large language models[J]. arXiv preprint arXiv:2106.09685, 2021.

 - About the literature:
    - Missing papers and comparisons with adaptor-based MTL methods. Adaptor-based MTL is very close to this paper, in which task-specific adaptors are parameter-efficient and plug-in modules on top of an all-shared feature extractor. For examples,
    [2] Rosenfeld, Amir, and John K. Tsotsos. "Incremental learning through deep adaptation." IEEE transactions on pattern analysis and machine intelligence 42.3 (2018): 651-663.
    [3] Zhao, Hanbin, et al. "What and where: Learn to plug adapters via nas for multidomain learning." IEEE Transactions on Neural Networks and Learning Systems 33.11 (2021): 6532-6544.

    - Missing recent papers in MTL/MDL.
    [4] Zhang, Lijun, et al. "Rethinking hard-parameter sharing in multi-domain learning." 2022 IEEE International Conference on Multimedia and Expo (ICME). IEEE, 2022.
    [5] Zhang, Lijun, Xiao Liu, and Hui Guan. "Automtl: A programming framework for automating efficient multi-task learning." Advances in Neural Information Processing Systems 35 (2022): 34216-34228.

 - About the experiments:
    - The authors may want to compare with adaptor-based MTL methods, which are the most related works to this paper.
    - It's necessary to have more experiments for MTL, including more baselines, more backbone models, and more datasets. More baselines could be the representative and new methods like MTAN and AutoMTL, rather than just RA and ASTMT. Backbone models could be some efficient backbone like MobileNet or EfficientNet, rather than just ResNet variants. Datasets could be CityScapes and Taskonomy, rather than just NYUD.
    - AdaShare is a typical MTL model. It's weird to compare it in MDL setting only.

**Questions:**

My main concerns are in the weakness part. Thanks.

**Limitations:**

The social impact is not mentioned either in the main paper or the supplementary materials.

---

> ### Author Rebuttal · Authors · 2023-08-10
>
> We would like to first thank the reviewer for the detailed analysis on our method and pointing out the concerns with missing references and experiments. We discuss all of them in detail below.
>
> 1. **Re method**
> - **FTN and MTL/MDL methods.**\
>       First, we would like to mention that many of the existing methods for MTL/MDL can be viewed as transfer learning or domain adaptation methods. The key question is how to learn multiple tasks/domains using parameter-efficient adaptation (either jointly or in an incremental manner). Our method also falls in this broad category.\
>       Second, we do **not** fine-tune the shared feature extractor for each task. We can view the updated weights as $W_t = W+\Delta W_t$, but we do not learn an arbitrary $W\_t$ and then perform low-rank factorization on $W\_t - W$ to reduce storage cost. We represent and optimize $\Delta W_t$ using low-rank factors, which is efficient in terms of parameter count and training.
> - **Advantages of using MTL** \
>       **Pretrained vs jointly-trained:** We agree that good parameter-sharing can improve the performance of MTL/MDL. Nevertheless, MTL/MDL methods can use a pretrained (frozen) network and add parameter-efficient adaptations (also called incremental MTL/MDL) or train a shared network using all the domains/tasks along with adaptations (also called joint MTL/MDL). For instance, LoRA method adds task-specific low-rank matrices to pretrained (frozen) weights; the Visual Decathlon dataset paper (Rebuffi et al. NeurIPS 2017) adds task-specific 1x1 filters and batch norm layers in a pretrained network; RCM paper also promotes incremental MTL approach.\
>       Two justifications often provided in favor of incremental MTL/MDL vs joint MTL/MDL are as follows. (1) Jointly training multiple tasks/domains can cause interference/negative transfer learning and cause performance degradation; (2) Incremental learning provides flexibility for adding new tasks/domains later when the corresponding data/labels become available (see additional discussion in TAPS and RCM papers).\
>       We included one experiment with joint MDL on DomainNet in Table 1 of supplementary, where FTN with rank-1 updates and joint MDL achieves better performance than incremental MDL with rank-40 (in Table 2, main paper). \
>       **Latency and cost.** We agree that our method does not provide savings in terms of computational cost and latency, and the same is true for most of the adaptation-based MTL methods. We will mention the latency and cost issue in the limitation section. To the best of our knowledge, Feature extractor-based methods with a shared backbone are the only methods that require one forward pass of input image (same encoder latency/cost as one task). We will be happy to add any other references you can provide for methods that avoid latency.
> - **Tensor vs matrix factors.** Tensor factorization can be viewed as a higher-order extension of PCA or SVD, and provides more diverse representation using fewer parameters. We will include some references. Since resnet weights are multi-dimensional tensors, tensor factor-based update is a natural choice. In contrast, to add low-rank matrices, we need to reshape filters into 2D arrays and then add low-rank factors. Note that FTN trained with full-rank ΔW is equivalent to fine-tuning the entire network (and equivalent to training an independent network per task/domain).
> - **Comparison with LoRA.** Thank you for pointing out this paper. Please see details in the Global and nJmA responses. FTN method provides better accuracy using fewer parameters compared to LoRA.
> 2. **Re related methods** \
> We will be happy to cite and discuss the papers you mentioned. For instance, "Incremental learning through deep adaptation" proposed deep adaptation networks (DAN) that add task-specific 1x1 convolutional layers and need nearly 13% new parameters per task. FTN requires only 2-6% new parameters. We compared FTN against PiggyBack, which outperforms DAN. RCM method also adds task-specific 1x1 convolutional layers.
> 3. **Re experiments.**
> - We believe experiments in our paper include comparison with adaptor-based MTL/MDL methods (e.g., series/parallel residual adapter on NYUv2). We will include comparisons with LoRA and KAdaptation (see table in Global response) and visual decathlon (xDWK response)
> -  During rebuttal, we performed an experiment for imagenet-to-sketch dataset with EfficientNet-b4 backbone. The table below shows the results, where FTN achieves similar accuracy as 6 Fine-tuned networks using ~4x fewer parameters.
>
> |                   |                |         |         |        |       |       |       |
> | ----------------- | -------------- | ------- | ------- | ------ | ----- | ----- | ----- |
> | Method            | Params         | Flowers | Wikiart | Sketch | Cars  | CUB   | mean  |
> | Fine-Tuning       | 6x(105.24M)    | 96.08   | 78.72   | 80.9   | 92.81 | 83.67 | 86.43 |
> | Feature extractor | 1x(17.54M)     | 80.91   | 42.37   | 56.3   | 42.97 | 64.77 | 57.46 |
> | FTN, R=1          | 1.079x(18.93M) | 93.28   | 74.07   | 79.77  | 87.93 | 82.82 | 83.57 |
> | FTN, R=10         | 1.474x(25.88M) | 94.79   | 77.54   | 80.7   | 89.71 | 84.70 | 85.48 |
> - We faced few issues in comparison with MTAN and AdaShare: \
> MTAN reports results on 13-class segmentation, whereas we report on 40-class segmentation. MTAN code also provided us better results on single task compared to what is in their paper. The difference in mIoU reported in Table 3 in MTAN vs Table 3 in our paper is huge. \
> AdaShare reports results for 3 tasks that are jointly trained. AdaShare parameter count is same as a single task network, but segmentation results are significantly poor compared to our method and other MTL methods we compared against. To avoid an unfair comparison, we did not include the results in our paper.
> - Due to limited time, we could not test cityscape/tiny-taskonomy but we will release our code for further evaluation.

---

> > ### Comment · Reviewer_TB3s · 2023-08-12
> >
> > Firstly, thank you very much for giving these detailed responses. Then let's talk about them in detail.
> >
> >  1. Yes, I fully understand that your method doesn't need to fine-tune the backbone. I'm trying to say that the process of fine-tuning the low-rank factorized tensors for each task is similar to fine-tuning from the same backbone weights and then doing the low-rank factorization on the incremented weights. You may want to include the comparison to show that your design is either trained faster or has better accuracy. That's why I feel this method is not very exciting to me - there is no new design on the adapter itself.
> >
> > 2. About the latency-benefit MTL methods, we may want to search for tree-structured multi-task models or branched multi-task models. Below are some references for you.
> > A tree-structured multi-task model recommender;
> > Learning to branch for multi-task learning;
> > Automated search for resource-efficient branched multi-task networks.
> >
> > 3. About the table you provide, I'm curious that if we do low-rank factorization on the 6 Fine-Tuning model's incremental weights, what will be its accuracy and number of parameters?

---

> > > ### Author Response · Authors · 2023-08-14
> > > **Follow up on low-rank approximation (part 1/3)**
> > >
> > > We thank you for your questions and apologize for taking some time in our response. We wanted to perform a thorough experiment to address your concern. We include several tables below, which caused the response to have multiple parts. We hope these results convince you that our design offers better accuracy and parameter efficiency. If you have any other questions or concerns, please let us know.
> > >
> > > ---
> > >
> > > **Re low-rank factorization of incremental weights (1 and 3).**
> > >
> > > If we understand your comments 1 and 3 correctly, you are asking what will happen if we compute the difference between weights of the fine-tuned network ($W_t$) and the original network ($W$), and then perform low-rank factorization on $\Delta W_t = W_t - W$ for all tasks and layers. We can view this as a compression or low-rank approximation of the trained $\Delta W_t$. Since an arbitrarily trained $\Delta W_t$ is not necessarily low rank, such an approximation can cause severe performance degradation.
> > >
> > > We performed a set of experiments to compare the performance of FTN against the low-rank approximation of weight difference in fine-tuned networks. We tested the following low-rank factorization methods:
> > >
> > > 1. CP decomposition that represents a tensor as a summation of rank-one factors. Note that FTN also represents $\Delta W_t$ as summation of rank-one factors, but those factors are learned during training.
> > > 2. Tensor train (TT) decomposition (also known as matrix product states) that often provides better approximation of tensors compared to CP using the same number of parameters.
> > > 3. SVD factorization of $\Delta W_t$ after reshaping every $C_{in} \times C_{out} \times k^2$ tensor into a $kC_{in} \times kC\_{out}$ matrix. *We do not claim this to be the best way to reshape the tensors into matrices, we are merely presenting the results for completeness.*
> > >
> > > We report the results below in terms of accuracy and parameter count for three factorization approaches with rank R = {1,10,20 30, 40, 50}. As R increases, we use more parameters to represent $\Delta W_t$, which reduces the approximation error (we also report all the approximation errors below).
> > >
> > > Our main finding is that low-rank approximation of trained $\Delta W_t$ causes severe performance degradation. For instance, R=1, average accuracy for imagene-to-sketch with ResNet 50 drops to nearly 27%. As R increases, the accuracy improves for all the factorization approaches, but they remain lower than FTN results.
> > >
> > > **Low-rank factorization of weight increments – Table 1- Imagenet-to-Text with ResNet-50 backbone**
> > >
> > > |  | | | | ||||
> > > | - | - | -| - | -| -| - | - |
> > > | Method | Params | Flowers   | Wikiart   | Sketch| Cars  | CUB   | mean  |
> > > | **Fine-Tuning (arbitrary ΔW_t = W_t -W)**  | **6x(105.24M)**| **96.08** | **78.72** | **80.9**  | **92.81** | **83.67** | **86.43** |
> > > | **FTN, R=1** | **1.004x(23.95M)** | **94.79** | **73.03** | **78.62** | **86.85** | **80.86** | **82.83** |
> > > | **FTN, R=50**  | **1.53x(36.02M)**  | **96.42** | **78.01** | **80.6**  | **90.83** | **82.96** | **85.76** |
> > > | ΔW with R=1 (CP)  | 1.009x(23.71M)  | 64.73 | 5.36  | 6.6   | 10.17 | 51.97 | 27.76 |
> > > | ΔW with R=10 (CP) | 1.09x(25.67M)  | 88.58 | 10.97 | 48.02 | 56.42 | 70.59 | 54.91 |
> > > | ΔW with R=20 (CP) | 1.185x(27.84M) | 92.76 | 20.83 | 61.88 | 78.65 | 76.98 | 66.22 |
> > > | ΔW with R=30 (CP)  | 1.27x(30.02M) | 94.15 | 37.68 | 71.27 | 83.56 | 79.81 | 73.29  |
> > > | ΔW with R=40 (CP) | 1.37x(32.197) | 94.6  | 46.33 | 74.73 | 86.74 | 81.2  | 76.72 |
> > > | ΔW with R=50 (CP) | 1.46x(34.37M)  | 94.75 | 56.65 | 77.32 | 87.95 | 82.03 | 79.74 |
> > > | ΔW with R=1 (TT) | 1.009x(23.71M) | 64.55 | 5.38  | 7.38  | 10.32 | 51.78 | 27.88 |
> > > | ΔW with R=10 (TT)  | 1.15x(27.18M)  | 88.4  | 8.87  | 52.33 | 57.64 | 70.14 | 55.47 |
> > > | ΔW with R=20 (TT) | 1.31x(30.86M)  | 92.73 | 22.26 | 62.7  | 78.34 | 77.01 | 66.60 |
> > > | ΔW with R=30 (TT) | 1.46x(34.53M)  | 94.13 | 37.66 | 70.9  | 84.16 | 80.22 | 73.41 |
> > > | ΔW with R=40 (TT)   | 1.62x(38.21M)  | 94.68 | 50.13 | 75.05 | 87.12 | 81.34 | 77.66 |
> > > | ΔW with R=50 (TT)   | 1.78x(41.89M)  | 94.89 | 57.24 | 77.55 | 88.17 | 82.24 | 80.01 |
> > > | ΔW with R=1 (SVD)   | 1.01x(23.79M)  | 62.69 | 6.19  | 5.88  | 12.21 | 50.21 | 27.43 |
> > > | ΔW with R=10 (SVD)   | 1.12x(26.42M)  | 88.75 | 12.3  | 52.00 | 54.21 | 71.26 | 55.70 |
> > > | ΔW with R=20 (SVD)  | 1.24x(29.34M)  | 92.81 | 20.89 | 63.35 | 77.63 | 77.03 | 66.34 |
> > > | ΔW with R=30 (SVD)  | 1.37x(32.26M)  | 94.18 | 35.2  | 69.77 | 83.65 | 79.48 | 72.45 |
> > > | ΔW with R=40 (SVD)  | 1.49x(35.18M)  | 94.55 | 51.04 | 74.38 | 86.64 | 80.96 | 77.51 |
> > > | ΔW with R=50 (SVD)  | 1.62x(38.13M)  | 94.89 | 58.92 | 76.45 | 87.82 | 81.71 | 79.95 |

---

> > > > ### Author Response · Authors · 2023-08-14
> > > > **Follow up on low-rank approximation (part 2/3)**
> > > >
> > > > The difference between FTN and post-training factorization is even more striking in the case of EfficientNet. For instance, R=1, average accuracy drops to nearly 5%. As R increases, the number of parameters increases sharply but even with 2x more parameters, the average accuracy of post-training low-rank factorization remains lower than FTN with R=1.
> > > >
> > > > **Low-rank factorization of weight increments -- Imagenet-to-Text with EfficientNet backbone**
> > > > | | | | |  | | | |
> > > > | -| - | - | - | - | - |- | - |
> > > > | Method | Params | Flowers   | Wikiart   | Sketch| Cars  | CUB   | mean  |
> > > > | **Fine-Tuning (arbitrary ΔW = W\_t -W)**  | **6x(105.24M)**| **96.08** | **78.72** | **80.9**  | **92.81** | **83.67** | **86.43** |
> > > > | **FTN, R=1**  | **1.079x(18.93M)** | **93.28** | **74.07** | **79.77** | **87.93** | **82.82** | **83.57** |
> > > > | **FTN, R=10**  | **1.474x(25.88M)** | **94.79** | **77.54** | **80.7**  | **89.71** | **84.70** | **85.48** |
> > > > | ΔW with R=1 (CP) | 1.04x(18.31M) | 20.08 | 0.65 | 0.4 | 0.83 | 3.02 | 4.99 |
> > > > | ΔW with R=10 (CP) | 1.43x(25.24M)  | 76.83 | 0.65  | 10.5  | 21.68 | 49.43  | 31.81 |
> > > > | ΔW with R=20 (CP) | 1.87x(32.95M)  | 91.72 | 0.65  | 46.63 | 72.73 | 73.23 | 56.99|
> > > > | ΔW with R=30 (CP)| 2.31x(40.66M)  | 94.68 | 2.32  | 68.53 | 85.59 | 79.27 | 65.07   |
> > > > | ΔW with R=40 (CP) | 2.75x(48.37M)  | 95.35 | 26.45  | 63.52 | 89.65 | 81.79 | 71.35 |
> > > > | ΔW with R=50 (CP) | 2.79x(49.02M)  | 95.45 | 41.42 | 76.5  | 91.03 | 82.34 | 77.34   |
> > > > | ΔW with R=1 (TT)  | 1.04x(18.31M) | 20.00 | 0.65  | 0.4   | 0.8   | 3.52  | 5.07  |
> > > > | ΔW with R=10 (TT)  | 1.43x(25.2M) | 77.02 | 0.65  | 11.17 | 22.62 | 49.19 | 32.13 |
> > > > | ΔW with R=20 (TT)  | 1.84x(32.3M)  | 91.49 | 0.66  | 47.5  | 72.94 | 72.51 | 57.02 |
> > > > | ΔW with R=30 (TT)  | 2.20x(38.76M)  | 94.78 | 2.37  | 68.38 | 86.08 | 79.46 | 66.21 |
> > > > | ΔW with R=40 (TT) | 2.52x(44.3M) | 95.32 | 31.64 | 74.6  | 89.79 | 81.95 | 74.66 |
> > > > | ΔW with R=50 (TT)   | 2.80x(49.22M)  | 95.63 | 63.18 | 78.5  | 91.07 | 82.84 | 82.24 |
> > > > | ΔW with R=1 (SVD) | 1.07x(18.77M)  | 23.74 | 0.65  | 0.4   | 1.02  | 6.02  | 6.366 |
> > > > | ΔW with R=10 (SVD)  | 1.51x(26.59M) | 76.84 | 0.65  | 11.33 | 22.81 | 49.43 | 32.21 |
> > > > | ΔW with R=20 (SVD) | 1.86x(32.68M)  | 91.59 | 0.66  | 47.62 | 72.96 | 72.52 | 57.07 |
> > > > | ΔW with R=30 (SVD) | 2.20x(38.63M)  | 94.78 | 2.37  | 68.38 | 86.08 | 79.46 | 66.21 |
> > > > | ΔW with R=40 (SVD)   | 2.51x(44.14M) | 95.32 | 31.64 | 74.6  | 89.75 | 81.95 | 74.65 |
> > > > | ΔW with R=50 (SVD) | 2.79x(49.09M)  | 95.63 | 63.18 | 78.5  | 91.07 | 82.84 | 82.24 |
> > > >
> > > > We also computed the approximation error (averaged over all the layers) for different values R and every task. We report average error and standard deviation over the layers for a subset of R. We observe that as R increases approximation error reduces, which leads to improvement in accuracy. Nevertheless, approximation error remains significant even with large R, which is the reason why accuracy remains significantly less than fine-tuned weights. In all cases, accuracy for tasks depends on approximation error (Flowers and CUBS do better than Wikiart and Sketch because the approximation error is small).
> > > >
> > > > **Average (std deviation)  approximation error for low-rank factorization of weight increments — Imagenet-to-Text with ResNet-50 backbone**
> > > >
> > > > First row shows norm of $\Delta W_t$ (averaged over all layers)
> > > > | | | | |  | | | |
> > > > | -| - | - | - | - | - |- | - |
> > > > | Method | Flowers | Wikiart | Sketch | Cars | CUB | mean |
> > > > | ΔW norm | 0.88 (1.18) | 31.15 (34.21) | 10.89 (11.95) | 2.23 (2.97) | 1.66 (2.54) | 9.36 (10.57) |
> > > > | ΔW with R=1 (CP) | 0.85 (1.15) | 30.56 (34.18) | 10.64 (11.93) | 2.16 (2.92)  | 1.60 (2.50) | 9.16 (10.53) |
> > > > | ΔW with R=10 (CP) | 0.63 (0.87) | 26.53 (31.00) | 9.07 (10.71)  | 1.72 (2.33)  | 1.27 (2.00) | 7.84 (9.38)  |
> > > > | ΔW with R=50 (CP) | 0.27 (0.36) | 16.20 (21.29) | 5.34 (7.02)   | 0.87 (1.16)  | 0.63 (0.93) | 4.66 (6.15)  |
> > > > | ΔW with R=1 (TT) | 0.85 (1.15) | 30.59 (34.20) | 10.66 (11.94) | 2.16 (2.92)  | 1.60 (2.50) | 9.17 (10.54) |
> > > > | ΔW with R=10 (TT) | 0.62 (0.87) | 25.90 (30.72) | 8.84 (10.57)  | 1.68 (2.31)  | 1.24 (1.99) | 7.65 (9.29)  |
> > > > | ΔW with R=50 (TT) | 0.25 (0.34) | 14.50 (19.81) | 4.78 (6.47)   | 0.77 (1.07)  | 0.56 (0.88) | 4.17 (5.71)  |
> > > > | ΔW with R=1 (SVD) | 0.85 (1.15) | 30.57 (34.21) | 10.65 (11.93) | 2.16 (2.92)  | 1.61 (2.50) | 9.16 (10.54) |
> > > > | ΔW with R=10 (SVD) | 0.65 (0.88) | 26.55 (31.23) | 9.11 (10.78)  | 1.74 (2.36)  | 1.29 (2.02) | 7.86 (9.45)  |
> > > > | ΔW with R=50 (SVD) | 0.29 (0.39) | 16.05 (21.69) | 5.38 (7.21)   | 0.89 (1.22)  | 0.65 (0.98) | 4.65 (6.29)  |

---

> > > > > ### Author Response · Authors · 2023-08-14
> > > > > **Follow up on low-rank approximation (part 3/3)**
> > > > >
> > > > > **Average (std deviation)  approximation error for low-rank factorization of weight increments –
> > > > > Imagenet-to-Text with EfficientNet backbone**
> > > > >
> > > > > First row shows norm of $\Delta W_t$ (averaged over all layers)
> > > > > | | | | |  | | | |
> > > > > | -| - | - | - | - | - |- | - |
> > > > > | Method | Flowers   | Wikiart   | Sketch| Cars  | CUB   | mean  |
> > > > > | ΔW norm | 38.78 (64.56) | 1115.38 (1920.09) | 360.35 (609.57) | 116.37 (189.08) | 94.03 (149.58) | 344.98 (586.57) |
> > > > > | ΔW with R=1 (CP)  | 36.33 (62.96) | 1054.37 (1858.15) | 339.09 (594.80) | 109.33 (185.27) | 88.60 (146.40) | 325.54 (569.51) |
> > > > > | ΔW with R=10 (CP) | 27.47 (52.43) | 788.24 (1496.99)  | 247.93 (488.69) | 82.10 (154.54)  | 67.01 (123.29) | 242.55 (463.18) |
> > > > > | ΔW with R=50 (CP)  | 16.42 (32.27) | 362.22 (668.03)   | 120.74 (251.63) | 45.45 (88.04)   | 38.23 (72.91)  | 116.61 (222.57) |
> > > > > | ΔW with R=1 (TT) | 36.34 (62.96) | 1054.40 (1858.14) | 339.11 (594.76) | 109.34 (185.27) | 88.61 (146.40) | 325.56 (569.50) |
> > > > > | ΔW with R=10 (TT)  | 27.47 (52.43) | 788.11 (1497.02)  | 247.86 (488.68) | 82.08 (154.54)  | 66.99 (123.29) | 242.50 (463.19) |
> > > > > | ΔW with R=50 (TT)  | 12.90 (29.40) | 287.13 (612.70)   | 96.42 (230.10)  | 35.79 (80.26)   | 30.05 (66.54)  | 92.45 (203.8)   |
> > > > > | ΔW with R=1 (SVD) | 36.01 (63.11) | 1045.52 (1862.29) | 335.92 (596.21) | 108.08 (185.84) | 87.69 (146.83) | 322.64 (570.85) |
> > > > > | ΔW with R=10 (SVD) | 27.28 (52.52) | 782.77 (1499.73)  | 245.92 (289.63) | 81.23 (154.97)  | 66.34 (123.63) | 240.70 (424.09) |
> > > > > | ΔW with R=50 (SVD) | 12.90 (29.40) | 287.13 (612.70)   | 96.42 (230.10)  | 35.79 (80.26)   | 30.05 (66.54)  | 92.45 (203.8)   |
> > > > >
> > > > > In summary, a fair comparison between FTN and post-training low-rank approximation would be to compare which method provides better performance for the same number of parameters. We observe that FTN clearly outperforms low-rank approximation of trained weight difference. The main reason is that fine-tuning the weights can cause arbitrary change in the weights and the difference is not compressible or low rank. In contrast, FTN enforces an explicit low-rank structure on the difference during training by learning a small number of factors, which provides parameter efficiency and accuracy.
> > > > >
> > > > > —
> > > > >
> > > > > **Re latency-efficient MTL (2).** Thank you very much for sharing these papers. We better understand your point about latency and computations. We will add the following discussion in the paper along references of the related papers
> > > > >
> > > > > Branched and tree-structured MTL methods enable different tasks to share branches along a tree structure for several layers. Multiple tasks can share computations and features in any layer only if they belong to the same branch in all the preceding layers. Such networks aim to reduce the computational cost and latency by keeping the number of branches as small as possible. In contrast, adaptation-based methods (including AdaShare, RCM, TAPS, and our method) can create as many paths as the number of tasks and need to compute or store features for each task separately. Such networks can compute features for all the tasks in parallel at the expense of a larger memory footprint.

---

> > > > > > ### Comment · Reviewer_TB3s · 2023-08-14
> > > > > >
> > > > > > Thank you for your efforts. That's clear to me now. I've raised my rate. Good luck.

---

> > > > > > > ### Author Response · Authors · 2023-08-14
> > > > > > > **Thank you for increasing the rating!**
> > > > > > >
> > > > > > > We appreciate all your comments and questions. Thank you for increasing your rating!

---

### Official Review · Reviewer_nJmA · 2023-07-01

**Soundness:** 3 good
**Presentation:** 3 good
**Contribution:** 2 fair
**Rating:** 6
**Confidence:** 4

**Summary:**

The paper presents the Factorized Tensor Network (FTN) as a solution to the challenge of learning multiple tasks/domains using a single unified network. The authors claim that FTN achieves comparable accuracy to independent single-task/domain networks with a smaller number of additional parameters. The method incorporates task/domain-specific low-rank tensor factors into a shared frozen backbone network. Experimental results on multi-domain and multi-task datasets demonstrate the effectiveness of FTN.

**Strengths:**

1. `Efficient Approach for Knowledge Transfer`: The Factorized Tensor Network (FTN) proposed in the paper offers an effective mechanism for transferring knowledge across multiple tasks and domains. By incorporating task/domain-specific low-rank tensor factors into a shared frozen backbone network, FTN allows for efficient utilization of existing knowledge, reducing the need for extensive retraining or independent models.

2. `Minimal Parameter Requirement`: FTN demonstrates the ability to achieve comparable accuracy to independent single-task/domain networks while requiring a smaller number of additional parameters. This parameter efficiency is advantageous when dealing with a large number of tasks or domains, as it helps reduce storage costs and computational complexity, making the approach more scalable and practical.

3. `Strong Validation Results`: The paper presents experimental results on widely used multi-domain and multi-task datasets. These results showcase the effectiveness of FTN, demonstrating similar accuracy compared to single-task/domain methods while utilizing only a small percentage (2-6%) of additional parameters per task. The validation experiments provide empirical evidence supporting the proposed approach and its potential for practical application.

**Weaknesses:**

1. `Limited Novelty`: The paper falls short in highlighting the novelty and distinctiveness of FTN compared to other parameter-efficient tuning methods, especially KAdaptation [A] and SSF [B] . This missed opportunity leaves readers wondering about the specific advantages and unique contributions of FTN. A more comprehensive analysis of its novelty would have evoked excitement and clarified its standout features.

2. `Limited Comparison`: The paper fails to sufficiently contrast FTN with other parameter-efficient fine-tuning (PEFT) methods, such as adaptors and low-rank approximation (LORA). This omission leaves unanswered questions about FTN's comparative strengths and weaknesses. A thorough evaluation against these methods would have added depth and fostered confidence in the proposed approach. Additionally, the paper's baseline for multi-domain image translation appears relatively weak, limiting the scope of the evaluation.

3. `Overlooking Recent Trends in Modular Learning`: The paper overlooks recent advancements in modular learning highlighted in references [C] and learning factorized knowledge with different modules [D].

[A] He X, Li C, Zhang P, et al. Parameter-Efficient Model Adaptation for Vision Transformers[C]//Proceedings of the AAAI Conference on Artificial Intelligence. 2023, 37(1): 817-825.

[B] Lian D, Zhou D, Feng J, et al. Scaling & shifting your features: A new baseline for efficient model tuning[J]. Advances in Neural Information Processing Systems, 2022, 35: 109-123.

[C] Pfeiffer J, Ruder S, Vulić I, et al. Modular deep learning[J]. arXiv preprint arXiv:2302.11529, 2023.

[D] Yang X, Ye J, Wang X. Factorizing knowledge in neural networks[C]//European Conference on Computer Vision. Cham: Springer Nature Switzerland, 2022: 73-91.

**Questions:**

The paper mentions that FTN achieves accuracy comparable to independent single-task/domain networks with a small number of additional parameters. Could you provide more insights into how FTN manages to strike this balance between accuracy and parameter efficiency? What are the underlying mechanisms or strategies employed by FTN that enable it to achieve such promising results?

**Limitations:**

The author should discuss some of the limitation of FTN in the paper

---

> ### Author Rebuttal · Authors · 2023-08-10
>
>
> We would like to first thank you for your detailed and insightful comments. We appreciate that you listed the strengths of our paper and pointed out the efficiency of our approach in terms of minimum parameter requirement without losing accuracy. We also find it interesting and appropriate that you characterize our work as a knowledge transfer approach.
>
> 1. **Regarding novelty and comparison with KAdaptation, LoRA, and SSF.** \
> Thank you for pointing out these papers. We mainly focused on ResNet architectures for a fair comparison with existing methods and regret missing a discussion of these papers (and other adaptation methods for transformers). To highlight the novelty and distinctiveness of FTN compared to parameter-efficient adaptation methods for transformer architectures, we will include references and a brief discussion of these papers in the related work section. During the rebuttal period, we also conducted preliminary experimental evaluations to compare the performance of our FTN approach against LoRA and KAdaptation (provided in the Global response above). We will provide detailed analysis in the main paper or supplementary material.
>
>    FTN shares some high-level similarities with other parameter-efficient adaptation methods such as LoRA, KAdaptation and SSF. All these approaches are aimed at introducing a small number of task/domain-specific parameters to adapt networks for multiple tasks/domains.
>
>    - LoRA is a low-rank adaptation method proposed for large language models, which freezes the pretrained weights of the model and learns low-rank updates for each transformer layer. In principle, any weight matrix can be updated with low-rank factors (e.g., LoRA updates weight matrices for query and value in every attention layer). Suppose the network has $L$ transformer layers with word embedding $d\_{model}$. The number of parameters needed to add rank r factors in query and value weight matrices is $2Lrd\_{model}$. \
>    - KAdaptation proposes a parameter-efficient adaptation method for vision transformers. KAdaptation proposes to represent the updates of MHSA layers using summation of Kronecker products $\sum\_i A\_i \otimes B\_i$. To further reduce the number of parameters, the $A\_i$ are shared across all layers while the $B\_i$ are represented as independent low rank factors for each layer. To the best of our understanding, the number of parameters required by KAdaptation is $2Lrd\_{model} + K^3$, where $K$ represents a design parameter in the Kronecker product. \
>    - Scaling and shifting your features (SSF) is another method for parameter-efficient adaptation that applies element-wise multiplication and addition to tokens after different operations $\gamma \odot x + \beta$. The number of parameters required by SSF is $mLd\_{model}$, where $m$ represents the number of SSF modules in each transformer layer.  SSF in principle is similar to fine-tuning the BathNormzation layer in convolutional layers, which has scaling and shifting trainable parameters. FTN trains the Batch Normalization layers and therefore has the capability to scale and shift features when adapting to new tasks.
>
>    FTN method proposes to add task/domain-specific low-rank tensors to shared weights. While we mainly present results for resnet architectures, low-rank tensors can be added to transformer layers as well. We have included details about comparison of FTN with LoRA and KAdaptation in the Global response above. Our main takeaway is that tensor factorization-based updates in vision transformers can achieve comparable accuracy to independent single-task/domain networks while requiring fewer additional parameters than existing parameter-efficient adaptation methods.
>
>    We used LoRA and KAdaptation codes and results in <https://github.com/eric-ai-lab/PEViT>. For instance, if we consider the results for updating the output projection weights of the MHSA layers. We reshaped the 768x768 output projection weight matrix into a 768 x 64 x 12 tensor ($d_{model}=768, d=64, n=12$) and added a rank-4 tensor per task/domain in each layer. Such an update provides mean accuracy better than LoRA and comparable to KAdaptation while using \~3.5x and \~2x fewer parameters compared to LoRA and KAdaptation, respectively.
>
>    We agree the image translation baseline is weak but it is mainly intended to show our method is applicable to generative tasks. We agree that more recent methods such as diffusion models would offer strong baselines. Even though we tried our best to cover a wide range of applications, we were unable to include such recent generative model based experiments in the current paper.
>
> 2. **Re overlooking recent trends in modular learning.**\
> We thank you for pointing out these papers. We will include the references in the related work section. The method proposed in \[D] uses a pretrained (multi-task) teacher network and decomposes it into multiple factor networks (each masters one specific task/knowledge) that are disentangled from one another. This factorization leads to sub-networks that can be fine tuned to downstream tasks, but they rely on knowledge transfer from a teacher network that is pretrained for multiple tasks.
>
> 3. **Re accuracy vs parameter efficiency.**\
> FTN based adaptation heavily relies on the idea that networks are over-parameterized and their objective space has low intrinsic dimension. FTN enforces low intrinsic dimension constraints by optimizing over low-rank tensors. We showed experimentally over a wide range of experiments that the adaptive landscape across different domains and tasks can in fact be optimized and approximated using low rank tensors.

---

> > ### Comment · Reviewer_nJmA · 2023-08-12
> > **Response to the author's rebuttal**
> >
> > I thank the author for the constructive feedback and the additional experiments provided to support the arguments. While I still have concerns about the novelty of the paper compared to other parameter-efficient tuning approaches, I think it's still valid that to bring
> > the idea of Module training for multi task and multi domain learning. i hope the the author can make the appropriate modification in the final version, and i raise my final rating to weak accept (6).

---

> > > ### Author Response · Authors · 2023-08-12
> > > **Thank you for your feedback and increasing the rating!**
> > >
> > > Thank you for acknowledging our efforts and increasing your rating!
> > > We thank you once again for all your suggestions and comments. They were super helpful.
> > > We will definitely include the additional experiments and comparisons presented in the rebuttal in the final version. We will also incorporate all of your and other reviewers suggestions.

---

### Official Review · Reviewer_yeXy · 2023-07-03

**Soundness:** 3 good
**Presentation:** 3 good
**Contribution:** 3 good
**Rating:** 6
**Confidence:** 4

**Summary:**

The paper tackles multi-task learning (MTL) and multi-domain learning (MDL). The author proposes to a general method to add task/domain-specific weights to the common shared backbone. Specifically, they add the factorized task/domain specific tensors which introduce very low storage cost for each task and domain while maintaining the original performance. They experiment in three different settings: multi-task learning with dense-prediction tasks, multi-domain learning for image classification and multi-domain learning for image generalization. They also ablate on the relationship between the rank of factorization and the accuracy.

**Strengths:**

(1) The paper is well written and easy to understand.

(2) The experiment section is well designed and they show the effectiveness of their proposed methods in three different scenarios including multi-task learning / multi-domain learning, deterministic and generative tasks.

(3) The design of the proposed method is reasonable, easy to implement but effective to preserve the performance of each task/domain and save the storage cost. It scales well with more tasks and domains

**Weaknesses:**

The paper is based on the convolutional network and the author uses big gan in the generation tasks. These designs are somehow in the old fashion. Even though it is mentioned in the conclusion that the method can be extended to transformer based architecture, it is lack of experimental proof.

**Questions:**

How does the model applied to the transformer based architecture?

Does the method work for diffusion models when tackling generative tasks?

**Limitations:**

They have included limitation in the main paper.

---

> ### Author Rebuttal · Authors · 2023-08-10
>
> We would like to first thank you for your detailed and insightful comments and we are glad that you were able to follow the paper easily. While we acknowledge that models like BigGAN may be considered old-fashioned, we believe they still serve as a good baseline for image generation tasks. Our main intention is to demonstrate that our proposed method is applicable to generative tasks, in addition to classification and dense prediction tasks.
>
> Your main questions are regarding the applicability of our proposed method to some recent architectures and models, namely transformers and diffusion models.
>
> **FTN applied to transformers.** We have applied our proposed method to transformers and conducted a set of experiments to show its effectiveness. We provided details in the Global response above. Our preliminary results suggest that FTN can be easily used to add low-rank factors in the MHSA projection matrices to achieve performance comparable to LoRA and KAdaptation using fewer additional parameters.
>
> **FTN applied to diffusion models.** We expect our proposed method to work on diffusion based generation tasks. This is mainly because it is agnostic to training objectives and the sampling method. Diffusion models commonly use U-Net based architecture and more recently in conjunction with transformers such as Diffusion Transformers (DiTs). We have experimentally shown our FTN method works with both architectures. Due to the limited rebuttal time and scope, we are unable to show experimental results specifically for diffusion models.

---

> > ### Comment · Reviewer_yeXy · 2023-08-14
> > **Response to the author rebuttal**
> >
> > I thank for the author's time and good presentation in the rebuttal. The authors addressed my first concern (applying this method to transformers) and I understand the time limit for the rebuttal so that the authors are unable to finish the diffusion model experiment. I am quite looking forward to see the results on diffusion model in the next version. And I will keep my score as 6 unchanged after the rebuttal.

---

> > > ### Author Response · Authors · 2023-08-14
> > > **Thank you for your response**
> > >
> > > We are glad that we addressed your concern about transformers.
> > > We will definitely test this approach with the diffusion models, but we are not sure if we will be able to add that in this paper.
> > >
> > > We really appreciate all your comments and how you summarized the strengths of our paper!
> > > Thank you for time, questions, and suggestion!

---

### Official Review · Reviewer_xDWK · 2023-07-07

**Soundness:** 3 good
**Presentation:** 3 good
**Contribution:** 2 fair
**Rating:** 6
**Confidence:** 3

**Summary:**

The paper tackles the problem of adapting a single pretrained network to multiple tasks or domains while introducing as little parameters as possible. More specifically, first a backbone model is trained which contains a shared backbone. Then,  the proposed method (Factorized Tensor Networks, **FTN**) is applied to further adapt the shared backbone to each task with a few added parameters. All task specific parameters are trained alongside each other (i.e. the BatchNorm parameter, the low-rank FTN parameters as well as the task heads)

Given a weight $W$ from the shared backbone, FTN adds low-rank parameters $\Delta W_t$ for each task/domain $t$ such that the forward pass for task $t$ uses the shifted weights $W + \Delta W_t$. Setting the rank $r$ of the low-rank parameters allow to control the trade-off between parameter efficiency and model capacity. In addition, FTN also tune batch norm layers parameters (scale and bias) for each task independently.

The proposed FTN is compared to previous methods on adapting weights during finetuning (e.g. residual adapters) on mainly three benchmarks:
- Multidomain classification from a pretrained ImageNet model to the 5 domains in ImageNet-to-Sketch
- Multidomain classification from a pretrained ImageNet model to the 6 domains in DomainNet
- Multitask from a pretrained ImageNet model to the 3 domains in NYU

**Strengths:**

* The proposed method is simple to implement and compared to other adapter methods it is extremely parameter efficient

* The method is experimented on both multi-domain (different inputs) and multi-task (different outputs) settings

* While the method is only evaluated on convolutional networks, there is no strong assumptions on the backbone architectures, hence it could also be used on transformers for instances

**Weaknesses:**

- **Missing important related work**: As far as I can tell, the proposed FTN is quite similar to the low-rank finetuning method from "LoRA: Low-Rank Adaptation of Large Language Models". While LoRA is only evaluated on NLP/transformers in the original paper, it should be mentioned and discussed in related work

- **Unclear how the number of parameters is computed** The original $BA^2$ paper reports a parameter cost of "1.03x" in their Table 3. While this paper reports a cost of "3.8x [1.71x]". It seems like the baselines results reported in this paper are directly taken from the Table 1 in the  "Task Adaptive Parameter Sharing for Multi-Task Learning" paer:   It would be nice to clarify how baselines evaluation was conducted and where the discrepancy may come from

- **Missing results from Visual Decathlon**: Manyworks on adapters report results on Visual Decathlon introduced in the original residual adapters work where an ImageNet pretrained model is adapted to a wide variety of very different downstream tasks

**Questions:**

* In **Table 3**: Why is the cost of "single task" equal to 4x the one of "decoder only" ? From Figure 1, it seems that "decoder only" would be an architecture contains one R18 backbone + 4 task heads. While "single task" would contain 4 R18 backbones, but still 4 decoders / task heads: So when comparing the absolute number of parameters (backbone + task head), shouldn't the factor be smaller ?


**Limitations:**

Limitations are explicitly mentioned in an independent paragraph and fairl ydescribe some drawbacks of the methods.
Although I partially disagree with the statement that *(line 73) "The proposed method does not affect the computational cost because we need to compute features for each task/domain using separate functional pathways"*: This is fine for the MDL applications, however most MTL works operate under the "feature -extractor" paradigm instead, where the multiple tasks can be solved with a single forward pass on the fully shared backbones, followed by lightweight task heads: In comparison, FTN is much less efficient as it requires individual forward passes of the encoder for each task.

---

> ### Author Rebuttal · Authors · 2023-08-10
>
> We would like to first thank you for your detailed and insightful comments and nicely summarizing the contributions and strengths of our paper. Below we provide our response to your comments and questions.
>
> ****
>
> 1. **Re missing related work (LoRA):** \
> Thank you for pointing out this paper. We will certainly add references and discussion of LoRA and other transformer-based papers in the related work section. As you pointed out, our method makes no strong assumptions on the backbone architectures, and can be used with transformers. Based on other reviewers comments, we also performed some experiments using transformer-based architectures during the rebuttal period. We will include them in the revised version of the paper as well. Our main takeaway is that using low-rank tensor updates with different transformer parameters can provide comparable or better results than LoRA with fewer parameters. We discuss LoRA and KAdaptation methods in detail in our global response.
>
> 2. **Re number of parameters for BA2:** \
> Yes, we used  baseline results reported in TAPS paper and reported the same numbers for consistency. TAPS paper reported numbers assuming that a boolean parameter is stored as 8-bits. They provided the following rationale in Sec. 4.1 under “Metrics”: “Methods \[..] that use a binary mask for their algorithm report the theoretical total number of bits (e.g., 32 for floats, 1 for boolean) required for storage rather than reporting the total number of parameters. However, as \[.] notes, depending on the hardware, the actual storage cost in memory may vary (e.g., booleans are usually encoded as 8-bits).” We will include this clarification in our paper or supplementary material.
>
> 3. **Re missing results with Visual Decathlon:** \
> We performed an experiment on Visual Decathlon dataset using resnet-18 backbone that was pretrained on Imagenet. We acknowledge that most of the papers report results for Visual Decathlon using resnet-26 backbone, but we were not able to perform a detailed analysis or reproduce the fine-tuning results using the code from the original Visual Decathlon repository.\
>    In the table below, we report the performance using FTN with rank R=1, 20 and comparison with Fine-tuning (one backbone per domain) and Feature-extractor (one shared backbone for all domains). We also report the total number of parameters used by each method. Our main findings are aligned with results reported in the main paper that FTN achieves accuracy significantly better than Feature extractor (and comparable to Fine-tuning networks) while using only \~6% additional parameters compared to Feature extractor.
>
>
> |                   |                |       |       |       |       |       |       |       |       |       |       |
> | ----------------- | -------------- | ----- | ----- | ----- | ----- | ----- | ----- | ----- | ----- | ----- | ----- |
> | Method            | # param        | Airc. | C100  | DPed  | DTD   | GTSR  | Flwr  | OGlt  | SVHN  | UCF   | mean  |
> | Fine-tuning       | 10x(111.7M)    | 49.74 | 80.08 | 99.91 | 53.19 | 99.93 | 84.60 | 88.37 | 95.43 | 80.63 | 81.32 |
> | Feature extractor | 1x(11.17M)     | 15.69 | 59.52 | 87.55 | 45.03 | 81.34 | 59.31 | 44.73 | 39.50 | 33.14 | 51.75 |
> | FTN, R=1          | 1.005x(11.23M) | 31.29 | 77.38 | 99.36 | 48.98 | 99.83 | 65.98 | 85.12 | 93.06 | 55.94 | 72.99 |
> | FTN, R=20         | 1.06x(11.89M)  | 42.93 | 77.41 | 99.82 | 52.97 | 99.94 | 76.96 | 86.12 | 94.6  | 65.52 | 77.36 |
>
>
> 4. **Re cost of single task vs decoder only:** \
>  Your observation is correct if we count all the parameters for the backbones/encoders and decoders/task heads. We are following the procedure adopted in other papers that only report the parameters for the backbone architecture. The ‘decoder only’ has one shared backbone, while the ‘single task’ has 4 independent backbones. Both of them have 4 separate decoders/task heads, the cost of which is not reported in Table 3. Each head is based on deeplabv3+, which uses approximately 18.49M parameters.
>
> 5. **Re limitations:** \
>  Thank you for this nice observation and correction. In the Limitation paragraph, we were mainly drawing a comparison of FTN with single-task/domain networks. We agree with your comment that in the case of MTL a shared backbone will require a single forward pass for all tasks while FTN will require as many forward passes as the number of tasks. In that sense, shared backbone will have lower computational cost compared to FTN. We will revise the paragraph to include this point.

---

> > ### Comment · Reviewer_xDWK · 2023-08-18
> > **Thanks for rebuttal clarifications**
> >
> > Hello authors,
> > thanks a lot for your reply and clarifications !
> > - The comparison to similar work on transformers was helpful
> > - The early results on Visual Decathlon are promising
> >
> > Overall I still have some reservations about the novelty of the proposed method due to existing related work, but the experimental evaluation is convincing and performed across a variety of settings, so I'll raise my score to 6 accordingly

---

> > > ### Author Response · Authors · 2023-08-18
> > > **Thank you for your response and increasing the score!**
> > >
> > > We thank you for acknowledging our efforts and increasing your score!
> > >
> > > We will include a discussion on novelty and significance of FTN compared to other methods based on all the experiments and analysis reported in the rebuttal.

---

### Author Rebuttal · Authors · 2023-08-10


We thank all the reviewers for their careful, insightful, and constructive comments. We very much appreciate that reviewers recognized our work as a simple and effective strategy that leads to significant improvements. Our proposed factorized tensor network (FTN) can achieve accuracy comparable to independent single-task/domain networks with a small number of additional parameters. Some reviewers characterized it as a parameter-efficient adaptation method for multi-task learning (MTL) and multi-domain learning (MDL), and we agree with that. The reviewers also acknowledged our attempt to cover a broad range of applications and tasks through well designed experiments and strong validation results.

We present our detailed responses to all of the individual comments under their respective reviews. We look forward to engaging in further discussions, and answering any further questions that the reviewers may have.
***
Below we discuss two main experiments we performed to answer some of the reviewers questions.

1. **Low-rank factorization methods for transformers.** Multiple reviewers expressed concerns regarding missing references and comparisons with adaptation methods for transformers (e.g., LoRA and KAdaptation). We have taken note of these concerns and conducted experiments by adapting our method to the transformer architecture. We discuss additional details and results below.

2. **Experiments with Visual Decathlon and EfficientNet backbone.** Furthermore, we have included experiments requested by reviewers for additional backbones and datasets. We present our findings on the visual decathlon dataset using the ResNet-18 backbone, as well as results on Imagenet-to-sketch transformation using the EfficientNet-B4 backbone. We provide additional details in the responses under reviewer comments.

**Discussion on adaptation with transformers.**

During the rebuttal period, in response to questions raised by several reviewers, we conducted some preliminary experiments to compare the performance of our FTN approach against LoRA and KAdaptation. We will provide detailed analysis in the main paper or supplementary material.

The key contribution of the FTN method is to add task/domain-specific low-rank tensors to shared weights. While we mainly present results for ResNet architectures in the main paper, low-rank tensors can be added to transformer layers as well.

We tested two such approaches
1. Following LoRA and KAdaptation, we can add low-rank tensor factors in the attention layers. For instance, we can represent all weight matrices for query or value in the Multi-Head-Self-Attention (MHSA) layer as a three-dimensional tensor of size $d_{model} \times d \times n$, where $d_{model}$ is the embedding dimension, $n$ is the number of heads and $d=d_{model} / n$. The total number of parameters needed for $r$ tensors in query and value weights at $L$ layers is $2Lr(d_{model}+n+d)$.
2. We can add low-rank factors in the output projection weights. The projection matrix can be represented as a three-dimensional tensor of size $d_{model} \times d \times n$ and adding $r$ rank-one factors will require $Lr(d_{model}+n+d)$ parameters.

In the table below, we present the results for supervised classification. As our base network, we use a pre-trained 12-layer pretrained ViT-B-224/32 (CLIP). We utilized the source code provided by KAdaptation to obtain the pre-trained weights and conducted experiments after extending the code base with our implementation of FTN. Results for Fine-tuning, Feature extractor, LoRA, and KAdaptation are taken from Table 3 of the KAdapation paper. The feature extractor method in our table corresponds to the Linear-probing method in the same table.

We ran experiments on CIFAR-10, CIFAR-100, DTD, and STL10 datasets. We report metrics specified in Table 8 of the KAdapation paper for each dataset. We also follow similar hyper-parameter tuning and training procedures. In the last two rows, we present the results of two variants of our proposed FTN method. In both experiments, we set the rank r=4. The first proposed method, FTN (query and value) surpasses LoRA in terms of average performance and requires fewer additional parameters. While it requires comparable parameters to KAdaptation, its performance is 0.8% lower. FTN (output projection only) requires ~2x fewer parameters than KAdaptation but achieves comparable performance. We believe KAdaptation is able to reduce the number of parameters due to weight sharing across layers. We did not explore such alternatives in our method due to time limitations. FTN was originally proposed for convolution based backbone networks and it is likely some of the hyper-parameter choices aren’t suitable for transformer architectures. Nevertheless, we have obtained very promising results that can be further improved.

|  |  |  |      |       | | | |
| :------------------------------: | :------: | :-------: | :--: | :---: | :----------: | :--------------: | :-------------------: |
| Method   | CIFAR-10 | CIFAR-100 |  DTD | STL10 | Avg accuracy | # total params\* | # additional params\* |
| Fine-tuning |   97.7   |    85.4   | 79.0 |  99.7 |     90.5     |   4x (351.6M)  |     4x 87,897,654     |
| Feature extractor  |   94.8   |    80.1   | 75.4 |  98.4 |     87.2     |    1x (87.9M)   | -  |
|  LoRA  |   95.1   |    78.1   | 78.1 |  99.2 |     87.6     | 1.006x (88.48M) |       4x 147,236      |
|  KAdaptation  |   95.9   |    84.8   | 78.1 |  99.2 |     89.5     | 1.003x (88.22M) |       4x 80,726       |
|     **FTN (Query and Value)**    |   95.8   |    83.4   | 77.1 |  98.7 |     88.8     | 1.003x (88.22M) |       4x 81,024       |
| **FTN (Output projection)** |   96.6   |    84.3   | 76.0 |  98.6 |     88.9     | 1.001x (88.0M) |       4x 40,512       |

*params do not include task/domain-specific classification heads.

*4x (351.6M) denotes Fine-tuning requires 4x base network parameters that is equal to 351.6M

---

### Decision · Program_Chairs · 2023-09-21

**Decision:**

Reject

**Comment:**

After reading the reviews and based on my own reading of the paper, I agree with the reviewer’s observation of very low-novelty and limited comparison with previous works. In particular, the paper focuses on tensor network (really just a tensor factorization of the domain/task specific learned weight). Yet there is no review or comparison with any of the related works, including some that are very similar [1, 2, 3, 4] and no references or theoretical explanation for the use of this factorization is given. All relevant papers should be at least discussed and at least the newest ones should be compared to, especially given how incremental the work is and how minor the differences are, beyond the choice of backbone architecture.

[1] A Unified Perspective on Multi-Domain and Multi-Task Learning, Yang and Hospedales, ICLR 2015
[2] Deep Multi-task Representation Learning: A Tensor Factorisation Approach, Yang and Hospedales, ICLR 2017
[3] Sharing residual units through collective tensor factorization to improve deep neural networks, Chen et al, IJCAI 2018
[4] Incremental multi-domain learning with network latent tensor factorization, Bulat et al, AAAI 2020

The lack of theoretical backing and lack of background on tensor methods and tensor networks alone are problematic but, more importantly, the lack of any mention or comparison with these very similar works is indispensable and I do not believe the paper can be accepted in the current state despite the thorough responses from the authors. In addition to the very related works above, I found there is a very large overlap with a previous paper that is not cited in the manuscript [5].

[5] Multi-Task and Multi-domain learning with tensor networks, Garg, Prater-Bennette and Asif, Proceedings of SPIE, 2023